# Structure of tetrameric forms of the serotonin-gated 5-HT3$_A$ receptor ion channel

Bianca Introini [1,2,10], Wenqiang Cui [3,4,10], Xiaofeng Chu[5,10], Yingyi Zhang[1,2], Ana Catarina Alves[6], Luise Eckhardt-Strelau[1], Sabrina Golusik[5], Menno Tol[6], Horst Vogel [3,6,7✉], Shuguang Yuan [3,8✉] & Mikhail Kudryashev [1,2,5,9✉]

## Abstract

**Multimeric membrane proteins are produced in the endoplasmic reticulum and transported to their target membranes which, for ion channels, is typically the plasma membrane. Despite the availability of many fully assembled channel structures, our understanding of assembly intermediates, multimer assembly mechanisms, and potential functions of non-standard assemblies is limited. We demonstrate that the pentameric ligand-gated serotonin 5-HT3A receptor (5-HT3AR) can assemble to tetrameric forms and report the structures of the tetramers in plasma membranes of cell-derived microvesicles and in membrane memetics using cryo-electron microscopy and tomography. The tetrameric structures have near-symmetric transmembrane domains, and asymmetric extracellular domains, and can bind serotonin molecules. Computer simulations, based on our cryo-EM structures, were used to decipher the assembly pathway of pentameric 5-HT3R and suggest a potential functional role for the tetrameric receptors.**

**Keywords** Pentameric Ligand-gated Ion Channels; Serotonin Receptors; Cryo-EM; Ion Channels
**Subject Category** Structural Biology

## Introduction

The formation of functional multimeric ion channels is a complex process that is not fully understood. Subunit composition, stoichiometry, and the relative positions of monomers within an oligomer are critical for the proper function of the mature ion channel. The synthesis, folding, and maturation of most eukaryotic cell proteins, including secretory and membrane proteins, occur at the endoplasmic reticulum (ER) membrane. Here, the nascent protein chain is transferred from the ribosome into the Sec61 translocation channel, which helps translocate hydrophilic polypeptide segments across the ER membrane and integrate hydrophobic transmembrane segments into the membrane (Lang et al, 2017; Rapoport et al, 2017; Itskanov and Park, 2023). Once in the ER, processing enzymes catalyze post-translational modifications, and ER-resident chaperones facilitate monomer folding and regulate their assembly into multimers (Wanamaker et al, 2003).

The assembly of multimeric ion channels, such as the pentameric ligand-gated ion channel (pLGIC) nicotinic acetylcholine receptors (nAChRs), is highly regulated. Only a small fraction of newly synthesized subunits adopt the correct folding and receive proper post-translational modifications necessary for oligomerization (Colombo et al, 2013; Wanamaker et al, 2003). For nAChRs, folding and oligomerization occur sequentially, with trimer intermediates detected minutes after subunit synthesis, followed by slower formation of tetramers and pentamers (Green and Claudio, 1993). Fully assembled channels leave the ER in vesicles that fuse with the trans-Golgi (Deutsch, 2003), undergo post-translational processing, and appear on the cell surface upon vesicle fusion with the plasma membrane. Another pLGIC, the serotonin receptor 5-HT3$_A$, has been assayed for its intracellular biogenesis, plasma membrane targeting, and ligand-induced internalization by time-lapse microscopy (Ilegems et al, 2004). Ligand-binding to newly synthesized 5-HT3$_A$R was detected at the ER three hours after transfection of HEK cells. Upon blocking the synthesis of new proteins, the 5-HT3$_A$R population in the ER and Golgi cisternae moved to the cell surface, indicating successful receptor folding and assembly (Ilegems et al, 2004). Some chaperones, like RIC-3, facilitate the maturation of acetylcholine receptors and 5-HT3$_A$R by interacting with the intracellular domains (Halevi et al, 2002; Pirayesh et al, 2020). Despite the characterization of the mature receptors, the rules that govern the assembly of pLGICs subunits into functional receptors and structural intermediates are not yet understood.

[1]Max Planck Institute of Biophysics, Frankfurt am Main, Germany. [2]Buchmann Institute for Molecular Life Sciences (BMLS), Goethe University of Frankfurt am Main, Frankfurt on Main, Germany. [3]The Research Center for Computer-aided Drug Discovery, Institute of Biomedicine and Biotechnology, The Shenzhen Institutes of Advanced Technology, Chinese Academy of Sciences, Shenzhen 518055, China. [4]University of Chinese Academy of Sciences, Beijing 100049, China. [5]Max-Delbrück-Center for Molecular Medicine in the Helmholtz Association (MDC), In Situ Structural Biology, Robert-Rössle-Str. 10, 13125 Berlin, Germany. [6]Institute of Chemical Sciences and Engineering (ISIC), Ecole Polytechnique Fédérale de Lausanne (EPFL), Lausanne, Switzerland. [7]Faculty of Pharmaceutical Sciences, Shenzhen University of Advanced Technology (SUAT), Shenzhen, China. [8]AlphaMol Science Ltd, Shenzhen 518055, China. [9]Institute of Medical Physics and Biophysics, Charité-Universitätsmedizin, Berlin, Germany. [10]These authors contributed equally: Bianca Introini, Wenqiang Cui, Xiaofeng Chu. ✉E-mail: horst.vogel@epfl.ch; shuguang.yuan@cadd2drug.org; mikhail.kudryashev@mdc-berlin.de

Studies suggest that 5-HT3$_A$R and other pLGIC family members are homo- or heteropentamers assembled from structurally homologous subunits (Wu et al, 2015). Five 5-HT3$_A$ subunits can form functional homo-pentamers (Kudryashev et al, 2016; Zhang et al, 2021; Polovinkin et al, 2018; Basak et al, 2018a; Hassaine et al, 2014), while 5-HT3$_A$ and 5-HT3$_B$ subunits can combine to form the hetero-pentameric 5-HT3$_{AB}$R with specific properties (Barrera et al, 2008). However, the details of the assembly process are unclear. Previous analyses suggested that 5-HT3$_A$R goes through different oligomeric states upon destabilizing events (Tol et al, 2013; Nicke et al, 2004; Boess et al, 1995). Recent cryo-EM structures of the pentameric 5-HT3$_A$R solubilized with detergent (Polovinkin et al, 2018; Basak et al, 2018a, 2018b, 2020) or reconstituted in saposins with lipids (Zhang et al, 2021) have been reported, but none revealed the oligomerization process leading to the mature pentameric form. High-resolution cryo-EM structures of the glycine receptor (Zhu and Gouaux, 2021) in detergent provided evidence that a pLGIC can undergo different assembly states, suggesting that tetramers are structural intermediates of this channel's assembly process. There are no studies that show such intermediate structures in native plasma membranes.

Here we report cryo-EM structures of tetrameric forms of 5-HT3$_A$R reconstituted into lipid bilayer nanodiscs and imaged in cell-derived plasma membrane vesicles. Both symmetrical and asymmetrical organizations of the receptor's extracellular domain were able to bind serotonin with different efficiency, but the binding did not open the tetrameric transmembrane ion pore. Cryo-electron tomography (cryo-ET) of plasma membrane vesicles resolved tetrameric and pentameric forms coexisting in the same membrane patches, strongly indicating the tetramers as intermediates in the receptor assembly process. To further investigate the mechanism of the assembly process of the 5-HT3$_A$R, we stepwise dissociated the pentameric receptor into lower oligomeric forms down to monomers while preserving the conformation of the subunits. Computer simulations modeled the stepwise pentameric receptor assembly, suggesting that incorporating a monomer into a tetramer is energetically more favorable than adding a dimer to a trimer. Our analysis resolves the diversity of oligomeric states of 5-HT3$_A$R in the membranes and contributes to the understanding of the receptor folding and assembly.

## Results

### Cryo-EM of 5-HT3$_A$R-Salipro reveals the presence of stable tetrameric forms

We purified the full-length murine 5-HT3$_A$R in a saposin lipoprotein-stabilized lipid bilayer and performed single particle cryo-EM, as we previously reported (Zhang et al, 2021). We imaged the receptor in the apo state and in the presence of serotonin, corresponding to the desensitized state (Appendix Fig. S1). 2D classification of the resulting particles surprisingly revealed that a fraction of proteins had tetrameric features (Fig. 1A). Consecutive 3D classification by CryoSPARC's heterogeneous refinement (Punjani et al, 2017) without imposition of symmetry revealed that while ~50% of particles adopted an expected previously described C5-symmetric arrangement (Zhang et al, 2021), other

particles unexpectedly formed a tetramer (Fig. 1C,E). Cryo-EM maps of tetrameric 5-HT3$_A$R in apo closed-state and serotonin-bound conformations were obtained at resolutions of 3.5 Å and 3.4 Å (Appendix Fig. S1).

Further classification of the tetrameric particles, without the application of symmetry, resulted in two structures: a near-C2-symmetric (which we will refer to as symmetric) and an asymmetric tetramer (Fig. 1B,D). The differences in the conformations were observed mostly in the extracellular domain (ECD): in the asymmetric tetramer, the ECD is organized similarly to the pentameric form of the receptor with one missing subunit (Fig. 1B), while the second conformation assembles as an apparent dimer of dimers (Fig. 1D). Both conformations showed tightly packed transmembrane domains (TMD) with a C4 symmetry between the membrane leaflets. In all the maps of the tetramers, the local resolution of the TMD reached 2.8 Å and the ECD around 3.6 Å, while the density of the intracellular domain (ICD) was poorly ordered due to the higher flexibility of this domain compared to the pentameric form of the channel (Fig. 1B,D; Appendix Figs. S2 and S3). In the apo-dataset 60% of the particles represent an asymmetric tetramer (Fig. 1B; Appendix Fig. S2) and 40% the symmetric one (Fig. 1D; Appendix Fig. S2) while in the serotonin-bound dataset, the ratio was 53:47 (Appendix Fig. S3). We observed densities resembling N-linked glycosylation in the extracellular domains of the tetramers in correspondence to N82, N164, and N148 as it was previously reported for pentameric 5-HT3$_A$R (Hassaine et al, 2014) (Fig. 1B,D, in red).

Treatment of 5-HT3$_A$R with SDS revealed progressive dissociation of pentamers into lower oligomeric states, with more monomers at higher SDS concentrations (Fig. 1F). Interestingly, the circular dichroism (CD) spectrum of the mostly monomeric fraction is nearly-identical to the untreated pentameric form (Fig. 1G). While our cryo-EM data demonstrates that the tetramers maintain the overall conformation of monomers, the unfolding experiments suggest that lower-order oligomers maintain a stable conformation even in aggressive environments.

Our cryo-EM maps allowed refinement of the available atomic model (PDB ID: 6Y59 (Zhang et al, 2021)) into the transmembrane and extracellular domains, but not into the intracellular domains. Comparison of the chains constituting the symmetric apo tetramer to the atomic model of the C5-symmetric apo-state (PDB ID: 6Y59 (Zhang et al, 2021)) showed a very similar structure of the domains with the average root mean squared deviation of the carbon atoms in the main chain (Cα RMSD) of 0.64 Å to 1.07 Å (Fig. 2A–C). The highest deviation of ~2 Å was detected in the helices M3 and in the generally more flexible horizontal MX helices (Fig. 2A).

The differences between the tetrameric conformations were larger in the extracellular domain (Fig. 2F). The extracellular domain of the chains C and D of the asymmetric tetramer are displaced by 6–14 Å outwards from the receptor's central axis, leaving a larger space between the extracellular domains of chains A and D (Fig. 2D–F). The transmembrane domains of the tetrameric conformations were less variable, with displacements of 1–4 Å (Fig. 2F,G). Given the new unexpected tetrameric forms, we wondered whether the tetramers are assembly intermediates for the pentamer or are they able to perform their own functions? To address the function of tetramers, we performed molecular calculations based on our structures.

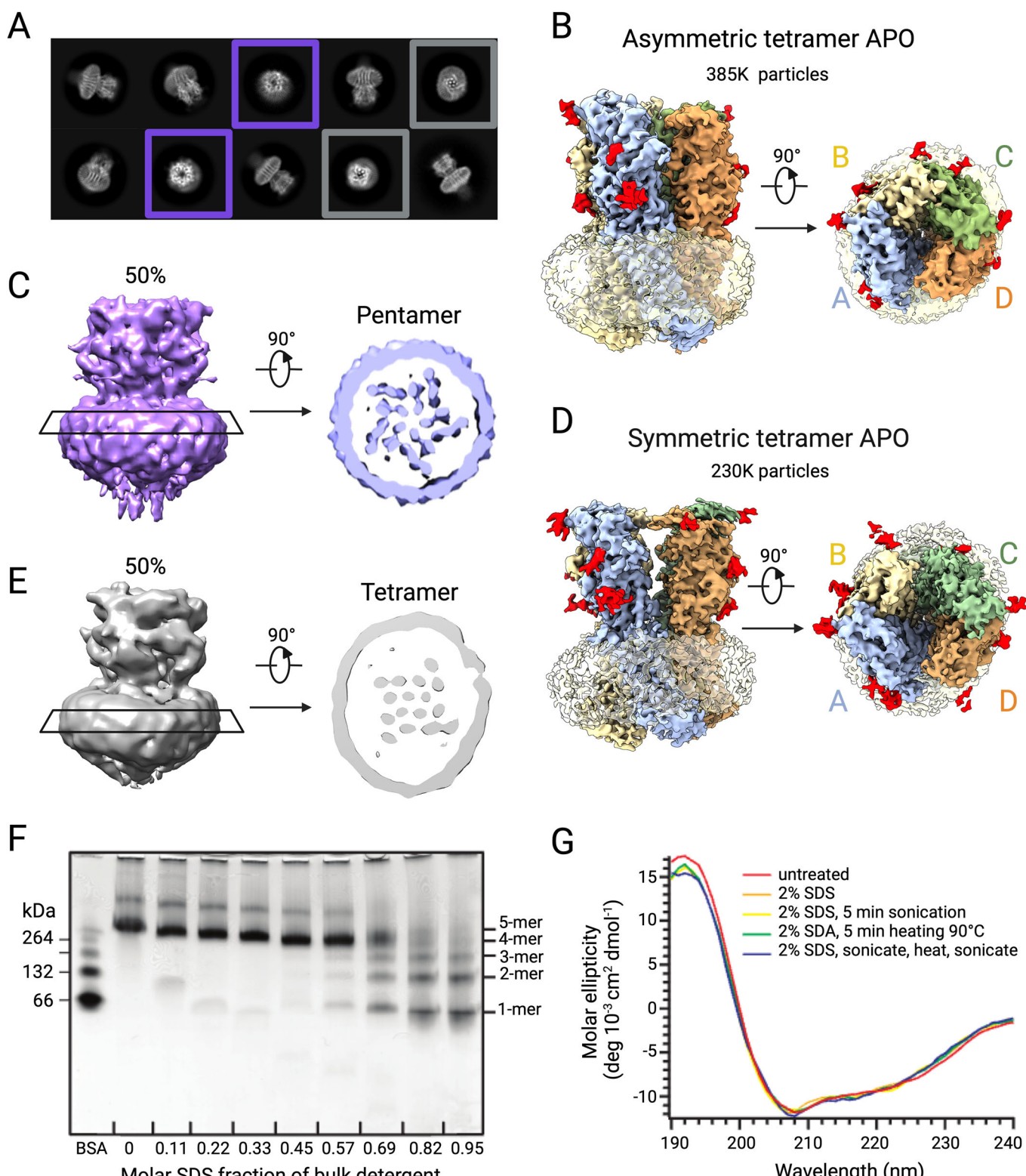

### Transition between the tetrameric forms of 5-HT3$_A$R

To assess whether the transition from one tetrameric form to another is energetically possible, we computed the translation free-energy surface (FES) expressed as Gibbs free energy (kJ/mol) at 310 K (Fig. 3A) using the "flexible axis" approach (Kutzner et al, 2011). The analysis of FES suggested that the transition from a symmetric (6.4 kJ/mol) to an asymmetric tetramer (3.6 kJ/mol) is

**Figure 1. Pentameric and lower oligomeric forms of 5-HT3_AR.**

(A) 2D class averages from the apo dataset of 5-HT3_AR in Salipro, pentameric (purple), and tetrameric (gray) arrangements. (B) Cryo-EM map of tetrameric 5-HT3_AR in an asymmetric conformation in the serotonin-free state. The subunits are color-coded: chain A is pale blue, chain B is pale yellow, chain C is green, and chain D is mustard. Salipro belt is in transparent yellow and N-glycosylations are in red. (C) Volume rendering of the apparent pentameric form of 5-HT3_AR, containing approximately half of the particles. Sections through the transmembrane domain shown on the right indicate the characteristic arrangement of the subunits in the TMD. (D) Cryo-EM map of tetrameric 5-HT3_AR in a symmetric conformation in the serotonin-free state. The subunits are color-coded: chain A is pale blue, chain B is pale yellow, chain C is green, and chain D is mustard. Salipro belt is in transparent yellow and N-glycosylations are in red. (E) Volume rendering of the apparent tetrameric form of 5-HT3_AR, containing approximately half of the particles. Sections through the transmembrane domain shown on the right indicate the tetrameric arrangement of the subunits in the TMD. (F) BN-PAGE of 5-HT3_AR unfolding by SDS. Samples were incubated for 30 min with increasing amounts of SDS (0 to 10 mM) and separated by native PAGE. Oligomers of BSA were used as size indicators (66, 132, 198, and 264 kDa). Oligomers of the receptor are indicated as 1-mer (monomer), 2-mer (dimer), 3-mer (trimer), 4-mer (tetramer), and 5-mer (pentamer). (G) Unfolding of 5-HT3_AR by SDS measured by CD spectroscopy. Samples containing 150 μg 5-HT3_AR were denatured by incubation with 2% SDS and a combination of sonication and heating, showing similar spectral profiles.

energetically possible and is likely to happen spontaneously ($\Delta G = -2.8\,kJ/mol \approx -0.67\,kcal/mol$). We next probed the transition of the symmetric to the asymmetric form. For this we used guided MD simulations, sampling the time interval of 1000 ps (Fig. 3B). Molecular dynamics (MD) simulations showed that the solvent-accessible surface area reduces during the transition from the symmetric conformation to the asymmetric one from 690 to 650 nm$^2$ (Fig. 3B). The reduction of the solvent-accessible area and of the gyration radius for the asymmetric state highlights that the asymmetric state is more compact than the symmetric one (Fig. 3A). During the progression from the symmetric to the asymmetric form we observed a redistribution of the numbers of hydrogen bonds between the chains with a decrease of the number of H-bonds between chains A-B and C-D and an increase between chains B-C (Fig. 3C).

Both the symmetric and asymmetric tetramers have a narrower pore compared to the previously reported pentameric structures (PDB ID: 6Y5A = asymmetric 5-HT-bound, PDB ID: 6Y59 = symmetric apo and PDB ID: 6Y5B = asymmetric apo). The range of pore width of 1.2–3.0 Å (Fig. 3D) suggests that water molecules should not penetrate the pore. We tested the ability of the tetrameric forms to hydrate the pore with all-atom molecular dynamics simulations and measured the number of water molecules in the pore. While few water molecules were observed close to both the intracellular and extracellular exits from the pore, the central ion pore region (corresponding to the residue L260) of both the symmetric and the asymmetric tetramers did not contain any water molecules during the whole long-time scaled MD simulations (Fig. 3E, top panels). This analysis suggests that the captured structures are not permeable to water and ions.

## Tetramers of the 5-HT3_AR are capable of binding serotonin

Our dataset of 5-HT3_AR, recorded in the presence of serotonin and without extracellular calcium, should correspond to a desensitized state of the receptor. Structural analysis revealed densities for serotonin in the ligand binding pockets (LBPs): for the symmetric tetramer two pockets were occupied—between the chains A and B (LBP1) and C and D (LBP3) (Appendix Fig. S4). The distance between the extracellular domains for the other chain pairs B/C and A/D was too wide to form an LBP (Fig. EV1A; Appendix Fig. S4D). For the asymmetric tetramer, two bound serotonin molecules were observed—between the chains A and B (LBP1) and between the chains B and C (LBP2) (Figs. EV3H–J and 4A,C–E). The interface

between the chains C/D at the extracellular side had a lower resolution (~4 Å) which was not sufficient to build an atomic model, therefore, we could not conclude if there is a serotonin molecule in the corresponding binding pockets. The distance between the chains D and A was too far to form a contact (Fig. 4A).

Upon serotonin binding, the receptor's LBP side chains compactify, the loop C "caps" the LBP, and side chains of W63, R65, Y126, W156, and Y207 interact with the serotonin molecule (Zhang et al, 2021; Polovinkin et al, 2018; Basak et al, 2018a). In tetramers, the displacement of the C-loop and of the interacting residues were minimal for the LBPs of the asymmetric tetramer and LBP1 of the symmetric tetramer. The serotonin poses were significantly different compared to fully functional pentamers (Figs. 4G,H and EV1G,H). Only for LBP2 of the asymmetric tetramer, the serotonin pose was similar to the fully assembled pentamer (Fig. 4H). Side chains in LBP2, specifically W63, W156, Y207, and F199, rearranged closer to the serotonin molecule, showing high similarity to the LBP of the functional pentamer (Fig. 4H; Appendix Fig. S4).

We evaluated the stability of the interaction between serotonin (5-HT) and the LBPs using the PDBePISA server (Krissinel and Henrick, 2007). We calculated the solvation-free energy ($\Delta iG$, kcal/mol) gain upon forming interfaces between a specific chain and the ligand, and the $\Delta iG$ $P$-value, an indicator of an interaction-specific interface (Figs. 4F and EV1F). $\Delta iG$ $P$-values < 0.5 indicate a specific molecular interaction due to higher hydrophobicity than expected for a non-specific interface. The previously reported C5-symmetric pentamer had an average $\Delta iG$ of $-3$ kcal/mol and a $\Delta iG$ $P$-value of 0.45. Similar values were only obtained for LBP2 of the asymmetric tetramer (LBP1 $\Delta iG$ $-2.7$ kcal/mol, $\Delta iG$ $P$-value 0.46, Fig. 4F,H). In contrast, the $\Delta iG$ for LBP1 of the asymmetric tetramer and both LBPs of the symmetric tetramer suggested less efficient binding (Figs. 4F,G and EV1F–H). Only for the asymmetric tetramer the calculated $\Delta iG$ $P$-value was significant ($P = 0.48$, Fig. 4F), suggesting a specific interaction. The $\Delta iG$ $P$-value calculated for LBP1 of the symmetric tetramer was >0.5 (i.e., 0.59, Fig. EV1F), suggesting a non-specific molecular interaction. No significant conformational changes in the transmembrane domains of the tetrameric forms were observed upon ligand binding (Figs. 4B and EV1B). Significant differences between the symmetric and asymmetric tetramer 5-HT-bound forms were identified primarily within the extracellular domain (Fig. EV2A–C). Chains A and B form the conserved LPB1, maintaining a stable conformation (Fig. EV2B,C). Notable alterations are observed within the extracellular domain of chains C and D (Fig. EV2B,C). The transmembrane domain exhibits a consistently stable arrangement across both serotonin-bound tetrameric forms (Fig. EV2D).

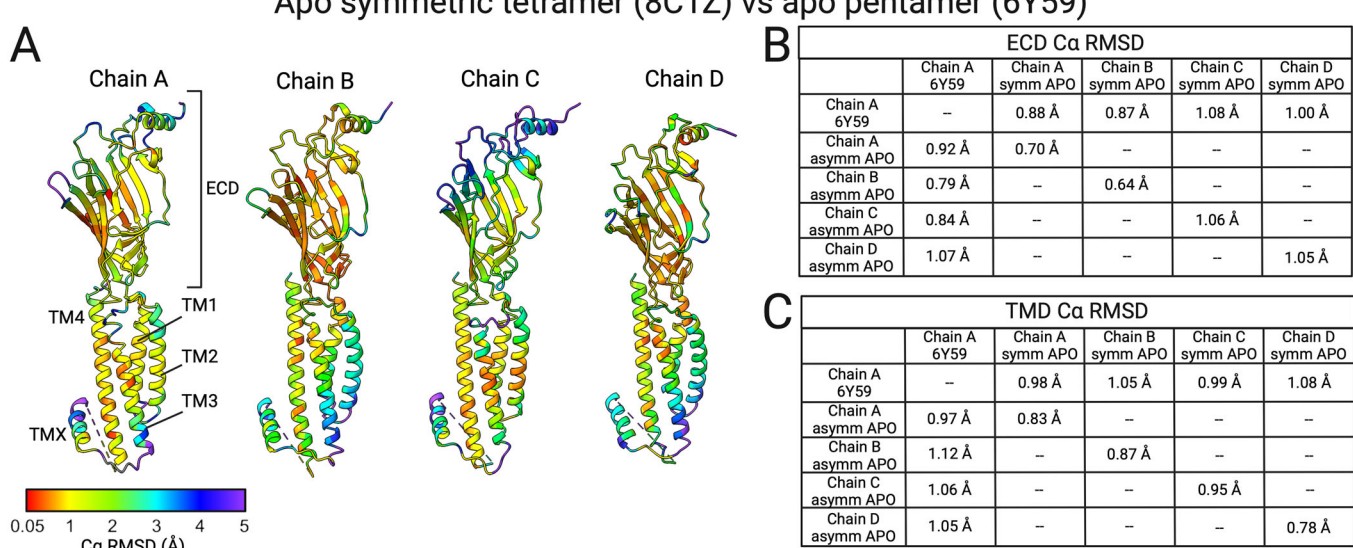

Apo symmetric tetramer (8C1Z) vs apo pentamer (6Y59)

### B — ECD Cα RMSD

| | Chain A 6Y59 | Chain A symm APO | Chain B symm APO | Chain C symm APO | Chain D symm APO |
|---|---|---|---|---|---|
| Chain A 6Y59 | – | 0.88 Å | 0.87 Å | 1.08 Å | 1.00 Å |
| Chain A asymm APO | 0.92 Å | 0.70 Å | – | – | – |
| Chain B asymm APO | 0.79 Å | – | 0.64 Å | – | – |
| Chain C asymm APO | 0.84 Å | – | – | 1.06 Å | – |
| Chain D asymm APO | 1.07 Å | – | – | – | 1.05 Å |

### C — TMD Cα RMSD

| | Chain A 6Y59 | Chain A symm APO | Chain B symm APO | Chain C symm APO | Chain D symm APO |
|---|---|---|---|---|---|
| Chain A 6Y59 | – | 0.98 Å | 1.05 Å | 0.99 Å | 1.08 Å |
| Chain A asymm APO | 0.97 Å | 0.83 Å | – | – | – |
| Chain B asymm APO | 1.12 Å | – | 0.87 Å | – | – |
| Chain C asymm APO | 1.06 Å | – | – | 0.95 Å | – |
| Chain D asymm APO | 1.05 Å | – | – | – | 0.78 Å |

Apo tetramer: from symmetric to asymmetric

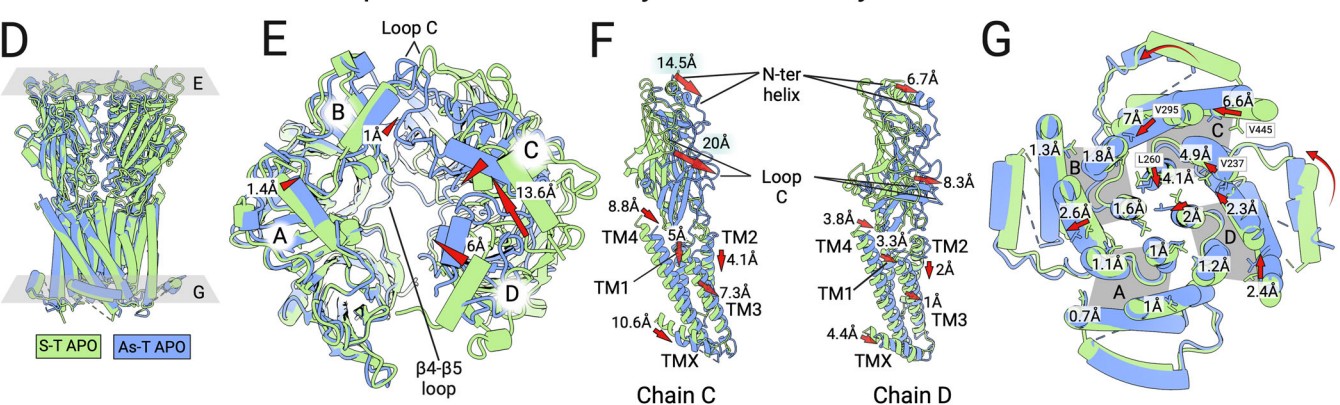

**Figure 2. Comparison of the protomers of tetrameric and pentameric forms of the 5-HT3AR.**

(A) Atomic models of single subunits of the tetramer, colored based on Cα RMSD value in Å when compared to a subunit of the pentameric apo C5 5-HT$_A$R (PDB ID: 6Y59). (B, C) Tables showing values of Cα RMSD (Å) at the level of the extracellular domain (ECD) and transmembrane domain (TMD) between the indicated chains. (D–G) Superposition of atomic models of the apo symmetric and asymmetric tetramers aligned at subunits A. (E) Top view of superposed symmetric (i.e., S-T APO) and asymmetric (i.e., As-T APO) tetramers. Red arrows indicate the differences between the N-terminal helices of the symmetric and the asymmetric structures. (F) Comparison of chains C and D of symmetric (sage green) and asymmetric (cornflower blue) apo structures when aligned with respect to subunit A. (G) Superposition of structures at L260 (9' position) of TM2. Displacements measured at the Cα atoms of indicated residues on each helix in the same cross-section are shown for (E–G). A downward displacement of both C and D helices is observed in the transition from symmetric to asymmetric. Source data are available online for this figure.

---

Structural analysis of the tetramers in the presence of serotonin and CaCl$_2$, which inhibits desensitization and allows the formation of the active state of the receptor, showed an overall resemblance to the ligand-unbound and -bound tetramers at 6 Å resolution (Fig. EV3). No noticeable displacements of the transmembrane helices were revealed, suggesting that the tetrameric receptors represent a closed ion pore (Fig. EV3E).

## Tetrameric forms are delivered to the plasma membrane where they co-localize with pentamers

We investigated whether tetrameric forms could be delivered to the plasma membrane, potentially playing a functional role. Using a previously described plasma membrane-derived microvesicle

system in human cell lines (Pick et al, 2005), we imaged 5-HT$_3$AR in microvesicles by cryo-ET. Tomograms contained distinguishable 5-HT$_3$AR-shaped densities (Fig. 5A–D). All observed 5-HT$_3$AR-containing vesicles in 309 tomograms showed the expected orientation when inserted in the plasma membrane. On the contrary, receptors localized at ER membranes would contain inverted protein topology (Duran and Meiler, 2013) and vesicles would occasionally contain ribosomes. The receptors were located in clusters, similar to previous findings in the plasma membrane of live HEK cells (Pick et al, 2003), with both pentameric and tetrameric receptors present (Fig. 5E). Examination of the inter-receptor distances indicated a clustering of pentamers in close proximity, while tetramers exhibited a more dispersed distribution with an average separation in the range of 30 nm

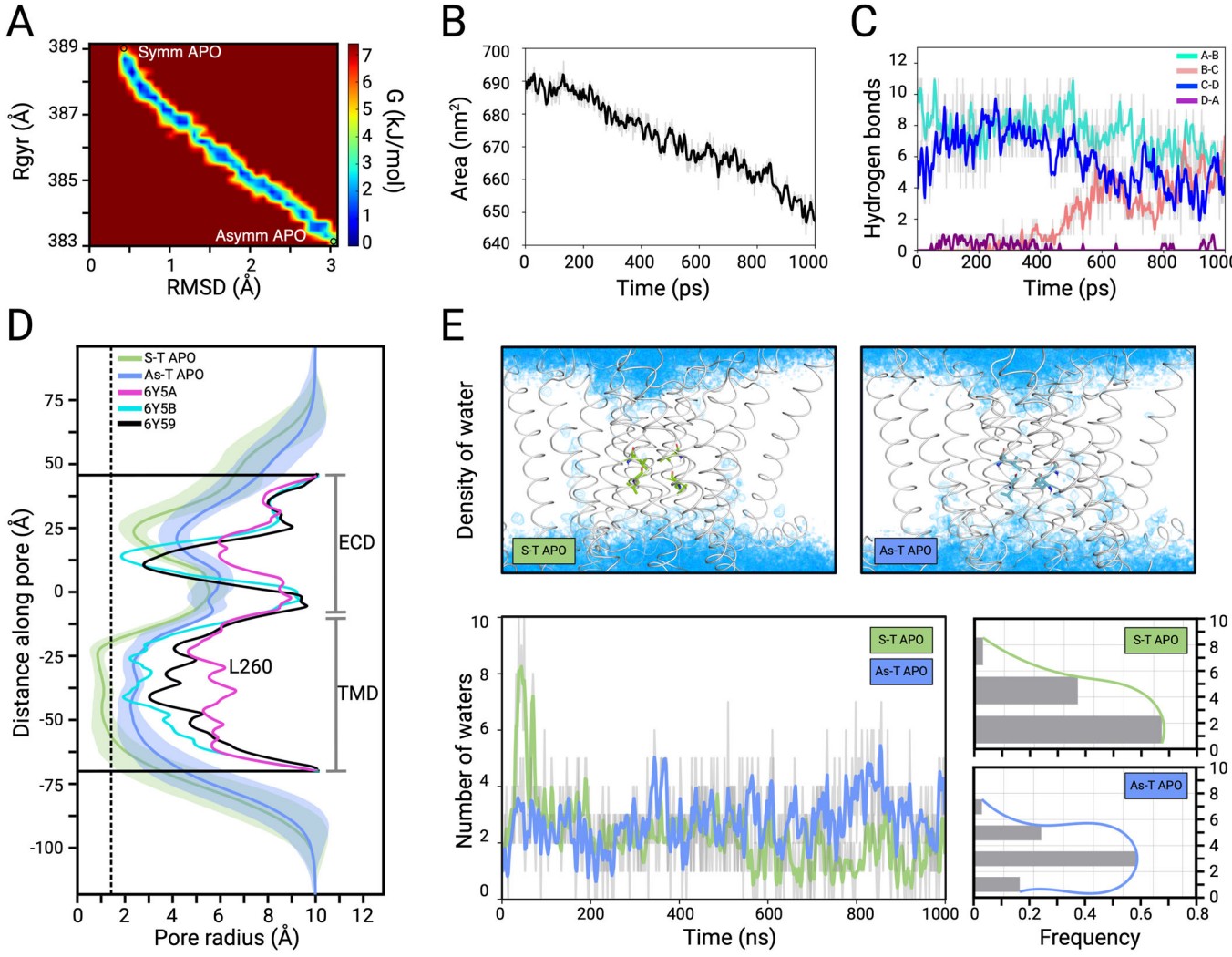

**Figure 3. MD simulations of the structural rearrangements from symmetrical to asymmetrical tetramer.**

(A) Free energy surface obtained from enforced rotation simulation (ERS) during the rearrangement, depicted as a function of RMSD (X-axis) and the radius of gyration (Rgyr) as the Y-axis. Gibbs free energy values (G, kJ/mol) are represented by a rainbow color code. (B) Changes in Solvent Accessible Surface Area (SASA) of tetramer structures over simulation time in the ERS. (C) Number of hydrogen bonds at the interface between two specific subunits during the transition from symmetric (time = 0 ps) to asymmetric tetramer (time = 1000 ps) in the ERS. (D) Pore radius for different structures. Solid lines represent the average radius over 1 μs conventional simulations, with the shaded area indicating the standard deviation. The vertical dashed line denotes the radius of a water molecule (1.4 Å). (E) Average water molecule density in the TMD within 4 Å from L260 during conventional MD. Top: Water molecule densities are shown in blue; the subunits of the indicated model are gray. For each subunit, the residue L260 is shown as sticks. Bottom: Number of water molecules observed for the apo tetramer symmetric (S-T APO, in sage green) and asymmetric (As-T APO, in cornflower blue)) over 1 μs simulations (left panel). The histograms (right panels) show the averaged distribution of the number of water molecules observed during the simulation on the left. Source data are available online for this figure.

(Fig. 5E). The tendency of pentamers to cluster aligns with previous observations in vivo within the plasma membrane of HEK cells (Pick et al, 2003).

We next performed subtomogram classification and averaging of the receptor-like densities without applying symmetry (Fig. EV4A–E). From the manually picked thirty thousand particles, a subset of 4431 particles generated a density map resembling the asymmetric tetramer observed in our single-particle analysis at a resolution of 25.2 Å (Figs. 5F,H and EV4E). Another subset of 3031 particles produced a density map depicting an apparently symmetrical pentameric receptor at a resolution of 19.6 Å (Figs. 5G–I and EV4D). Despite more particles contributing to the tetrameric isoform, the resolution was

higher for the pentamer, suggesting the tetramer is more flexible. Both atomic models of tetramers from our single-particle cryo-EM analysis fitted well into the tetrameric map from subtomogram averaging (Fig. 5H).

## Energy landscapes and assembly of multimers

We next computationally probed if the tetrameric forms of 5-HT3AR could serve as intermediates for the assembly process of the pentameric receptors. For this, we used coarse-grained atomistic MD simulations in combination with metadynamics simulations, a widely used method for the sampling of rare biological events by analyzing

## Only the extracellular domain reorganizes upon 5-HT binding

**Figure 4. The binding of serotonin does not lead to conformational changes in the transmembrane domain.**

(A, B) Superposition of the apo- (i.e., As-T APO) and serotonin-bound (i.e., As-T 5-HT) structures for the asymmetric tetramer. (A) Red arrows indicate the direction of the movement of the N-terminal helices. As we observed for the symmetric form, the HOLO conformation showed densities for serotonin only in two LBPs (numbered 1 and 2). (B) Cross-sections at the TMD residue L260 (9′ position) of M2. Displacements measured at the Cα atoms of the indicated residues on each helix in the same cross-section are shown. No large movements were observed. (C, D) Close-up of the LBPs of the apo and serotonin-bound tetramers. Serotonin is shown in orange. (E) Depicts the planes shown in (A) and (B) and serotonin densities are shown in orange. (F) Solvation-free energy gain (ΔiG, kcal/mol) on formation of the interface between indicated chains and ligands. Negative values indicate a hydrophobic interface, thus the more negative the value, the stronger the interaction. The ΔiG P-value is defined in (Krissinel, 2009). It is a measure of the specificity of the interface, P < 0.5 indicates interfaces with a higher-than-expected hydrophobicity, suggesting a specific molecular interaction. ΔiG and ΔiG P-values were calculated using the PDBePISA server (Krissinel and Henrick, 2007). PDB IDs of each tetrameric form deposited is reported above each panel. (G, H) Superpositions of the LBPs occupied by serotonin of the HOLO tetrameric 5-HT3R (As-T 5-HT) with the respective LBPs of the pentameric 5-HT3R (PDB ID:6Y5A). Serotonin molecules bound to the tetramers are displayed in orange while those bound to 6Y5A are in lavender

free energy landscapes of biomolecules (Barducci et al, 2011, 2008; Bussi and Laio, 2020). Starting from monomers, we simulated the possible oligomerization steps that can lead to the formation of a final pentameric 5-HT$_3$AR (i.e., Monomer + Monomer = Dimer, Monomer + Dimer = Trimer, Trimer + Monomer = Tetramer, and so on, as shown in Fig. EV5A–F). We calculated free energy surfaces (FES) for given oligomerization states as a function of distance between extracellular regions and transmembrane regions (Fig. EV5A–F). FES analysis for the interaction between two monomers and between a dimer and a monomer, showed energy barriers of 10.2 and 6.9 kcal/mol respectively (Figs. 6 and EV5A, B). Two scenarios for tetramer formation are possible: a symmetric tetramer forms from two dimers with a barrier of 16.8 kcal/mol, or a potentially less symmetric tetramer forms from a trimer and a monomer with a barrier of 12.3 kcal/mol

(Figs. 6 and EV5C,D). Overall, the global minimum of FES appeared at low distances between the subunits (ECD < 50 Å and TMD < 40 Å) for all the calculations, suggesting that the formation of lower oligomeric states is energetically favorable.

Interestingly, there were alternative local minima in the FES plots for the formation of tetramers and pentamers (Fig. EV5D,E). The FES for the formation of the tetramer from a trimer and a monomer shows two energy minima of −33.6 and −33.5 kcal/mol (Fig. EV5D, white and red circles). The structure of the symmetric tetramer observed in our cryo-EM data (Fig. EV5D, black circle) is a few Angstroms away from both those energy minima. The formation of pentamers from trimers and dimers also has two minima of FES which are further away from each other (Fig. EV5E, white and red circles). The minima closer to the observed structure

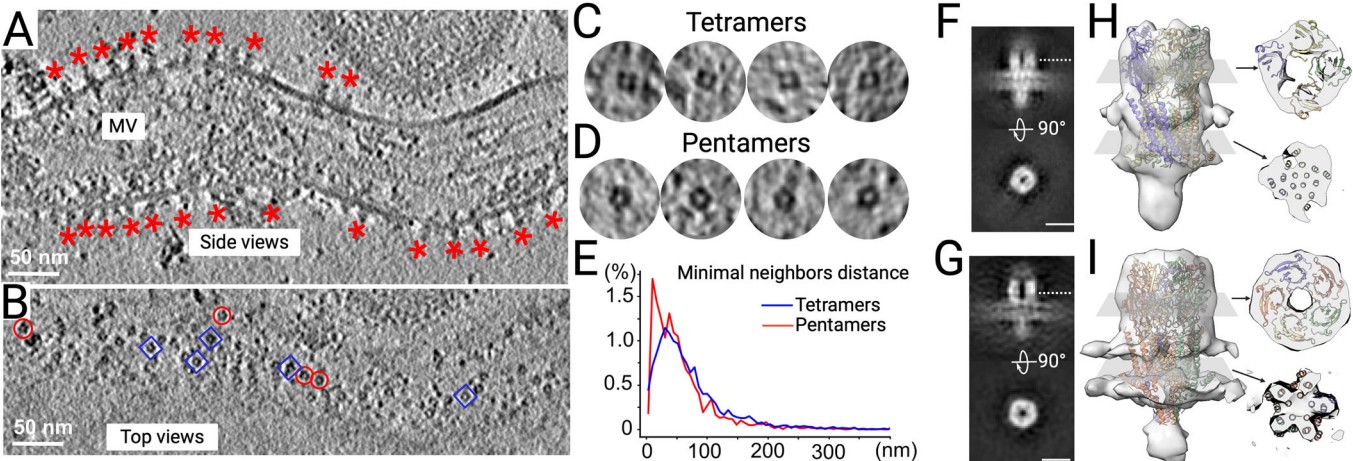

**Figure 5.  Structural analysis of tetrameric and pentameric forms of 5-HT3$_A$R in plasma membrane vesicles.**

(A, B) Slices through tomograms of native microvesicles (MV) from cells overexpressing 5-HT3$_A$R. Scale bars: 50 nm. (A) Side views of the individual receptors are marked with red asterisks. (B) Top views of the receptors in the same vesicle; red circles highlight pentamers, blue squares—tetramers. (C, D) Views through individual tetramers (C) and pentamers (D). (E) Distance between closest neighbors in plasma membrane-derived microvesicles. (F, G) Subtomogram average structures of a tetramer (F) and a pentamer (G). The dotted white line indicates the slice of extracellular domain shown in (H) and (I). Scale bars are in solid white: 10 nm. (H, I) Volume-rendered visualizations of the average structures in native microvesicles overlaid with the atomic models of the asymmetric tetramer (H) and of the pentamer (I, PDB ID: 6Y59). Source data are available online for this figure.

(Fig. EV5D, black circle) corresponds to the energy minima of −46.2 kcal/mol. The major differences observed between the two alternative energy minima were observed in the ECD domain (Appendix Fig. S5). The energy barrier for the formation of a pentamer from a tetramer and a monomer is 11.1 kcal/mol (Figs. 6 and EV5F) and it is lower than the addition of a dimer to a trimer. Therefore, from the energy landscape point of view, the lowest energy valley for the assembly of a pentamer is the addition of monomers to the existing complex one by one (Fig. 6).

We next conducted guided molecular dynamics simulations to form oligomers from specific combinations of precursors. For this, we computationally pulled away one subunit from a particular assembly over a simulation time of 500 ns. Reversing this process on the time axis corresponds to the formation of a particular oligomer of the receptor starting by forming a dimer from the interaction of two monomers and then proceeding with the formation of higher-order oligomers (Fig. 7A,B). During the formation of different oligomeric forms, the initial contact area was similar across all simulations: the hydrophobic TM domains of the interacting subunits approached each other first, followed by a structural rearrangement of the ECD region (Fig. 7C–H). In dimer formation, the first contact between the two monomers occurred over the entire length of their M2 helices (Fig. 7C). For higher-order oligomers, the first contact between interacting subunits was at the inner leaflet of the membrane (Fig. 7D–H). A simulation of pentamer formation from an asymmetric tetramer is presented in Movie EV1.

## Discussion

We demonstrated that the homo-pentameric ligand-gated ion channel 5-HT3$_A$R can form stable tetramers in lipid bilayers and cellular membranes. In the presence of SDS, fully assembled

pentamers dissociate into lower-order oligomers down to monomers, preserving the overall conformation of the subunits. However, as we estimate, this process would have a high energetic cost, which makes dissociation of pentamers during sample preparation or in natural cellular conditions unlikely. It is therefore tempting to attribute the tetrameric structures to assembly intermediates of the receptors. Glycine receptors from the cys-loop pLGIC family were recently purified from brain spinal cord in a trimeric and a tetrameric form, suggesting that these forms represent partially assembled alpha subunits of a hetero-pentameric receptor (Zhu and Gouaux, 2021). Unlike the glycine receptor tetramers, which exhibit an unfilled space in the transmembrane domain, our tetramers form a nearly C4-symmetric arrangement in the TMD, suggesting a more stable configuration. Lower oligomeric forms of 5-HT3$_A$R may also be present in our cryo-EM samples, but at lower amounts, preventing their reliable detection by this technique. Furthermore, we demonstrate not only the existence of tetrameric forms but also their successful delivery to the cellular plasma membrane. This suggests that tetramers passed the quality control mechanisms for membrane protein assembly. Interestingly, the presence of tetramers in a purified preparation for single-particle cryo-EM was recently reported (López-Sánchez et al, 2024), however, their fraction was low, which did not allow obtaining high-resolution structures. In our current work, as well as in the report by López-Sánchez and colleagues, the protein was produced in the tetracycline-inducible expression system in HEK293 cells, optimized for high protein yield, resulting in up to $10^7$ receptors per cell (Hassaine et al, 2013). This is higher than the number of 5-HT3$_A$R per neuronal cell, therefore the ratio of the tetramers to pentamers in vivo might differ.

Despite the high structural similarity between the individual monomers in the tetrameric and pentameric forms of the receptor, the inter-protomer interactions, as well as the poses of serotonin in

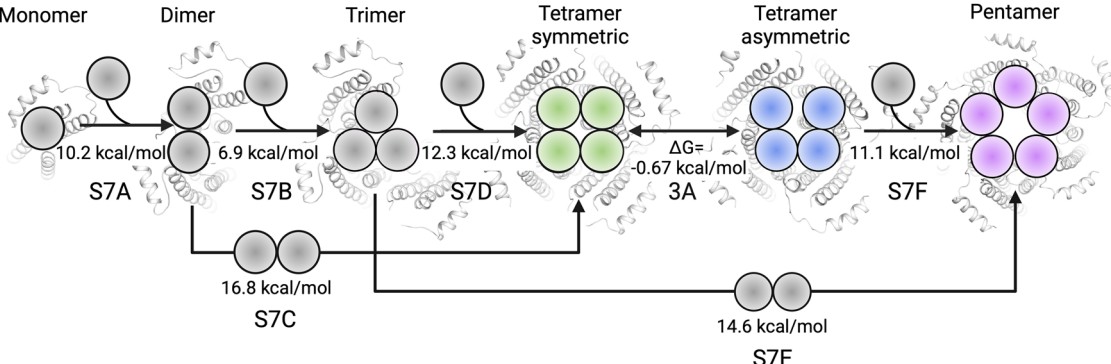

**Figure 6. Proposed model for pentamer assembly pathway.**

The energy barrier values necessary to transit from one oligomeric state to another are expressed in kcal/mol. Under each arrow S7A–F refers to one of the FES representations shown in Fig. EV5A–F and 3A refers to Fig. 3A.

the ligand-binding pockets are different. The only ligand binding pocket that showed the pose of the serotonin molecule similar to a functional pentamer was the LBP2 of the asymmetric tetramer. LBP2 is located at the sole interface where each of the ECD domains has neighboring domains that seemingly stabilize the LBP, facilitating the efficient binding of serotonin. Therefore, our data suggests that for serotonin to bind functionally to an LBP, both 5-HT3R monomers forming the LBP have ordered neighboring subunits. However, as there is only one LBP resembling the pentameric receptor, the tetrameric receptors have a closed pore, a conclusion that we confirmed by MD simulations. In our cryo-EM structures, ligand binding also did not lead to functional displacement of the transmembrane helices. Previous single-channel analysis of α7-nAChR/5-HT3R chimeras demonstrated that the generation of stable currents required the binding of two to three ligands in non-consecutive binding pockets (Eiselé et al, 1993; Rayes et al, 2005). The absence of the functional consecutive LBPs in tetramers prevents the receptors from opening in response to serotonin binding.

Our simulations suggest an assembly mechanism for 5-HT3AR: two monomers combine to form a dimer, which subsequently assembles into a trimer with the addition of another monomer. This trimer then incorporates another monomer to form a tetramer. Tetramers exist in two states: symmetric and asymmetric, with the ability to interchange between them with low-energy barriers. Finally, a pentamer is formed by adding a monomer to the asymmetric tetramer. The assembly process of 5-HT3AR multimers initiates with interactions among the transmembrane domains, followed by associations between the extracellular domains. While 5-HT3ARs and some other pentameric ligand-gated ion channels (e.g., nAChR) feature an intracellular domain that engages with transmembrane chaperones such as RIC-3 during receptor assembly, our data did not reveal the presence of associated chaperones. This absence was noted both in purified receptors and in cellular membranes. However, RIC-3 is not essential for the 5-HT3R assembly but enhances the surface expression of the receptors and modulates the subunit composition (Walstab et al, 2010; Cheng et al, 2005). The ability of the extracellular domains of pLGICs to form pentamers independently of the transmembrane domain is well-established. Our simulations showed that

transitioning from an asymmetric to a symmetric tetramer requires surmounting a minor energy barrier ($\Delta G = -0.67$ kcal/mol). Computational pulling experiments further demonstrated that the transmembrane domains are pulled apart right after the extracellular domains, suggesting that the initial interactions are primarily driven by hydrophobic interactions within the TM domains. At the cytoplasmic side of the 5-HT3AR membrane the receptor harbors short horizontal MX-helices positioned adjacent to the M4 helices, while the interactions between the protomers are modulated by M1, M2, and M3 helices. We previously suggested that the MX helices mediate interactions between assembled receptors in membranes (Kudryashev et al, 2016). Given that most pLGICs (e.g., GABAAR, GlyR), including bacterial counterparts, lack intracellular domains or MX helices (Jaiteh et al, 2016), we propose that the assembly mechanism of oligomers from precursors, driven by hydrophobic forces, could be conserved. However, the specific functions of intracellular domains and MX helices may vary depending on the protein.

Tetrameric forms of 5-HT3AR, which may represent assembly intermediates preceding pentamers, are delivered to the plasma membrane, suggesting possible functional implications. A related pLGIC, α7 nAChR, mediates calcium signaling via both ionotropic and metabotropic mechanisms (Kabbani and Nichols, 2018). Due to the similarities in the overall architecture of 5-HT3AR and α7 nAChR, it is possible that tetrameric 5-HT3AR could also interact with other molecules and perform signaling functions without ion flow through the pore. Moreover, preclinical studies of 5-HT3AR antagonists in depression and comorbid anxiety models showed encouraging results (Bhatt et al, 2013; Gupta et al, 2011). Competitive antagonists of 5-HT3R were proposed to increase the presence of free 5-HT, thereby enabling its binding to other serotonin receptors (Bhatt et al, 2021). Thus, non-ionotropically active 5-HT3AR tetramers could contribute to serotonin homeostasis in the brain and regulate the availability of 5-HT. Furthermore, we observed the presence of microdomains in plasma membrane-derived vesicles characterized by the presence of densely packed tetrameric and pentameric receptors. These formations may arise from interactions among receptors or between specific lipids and receptors. Elevated concentrations of receptors in localized areas could enhance the efficiency of ligand binding and potentially

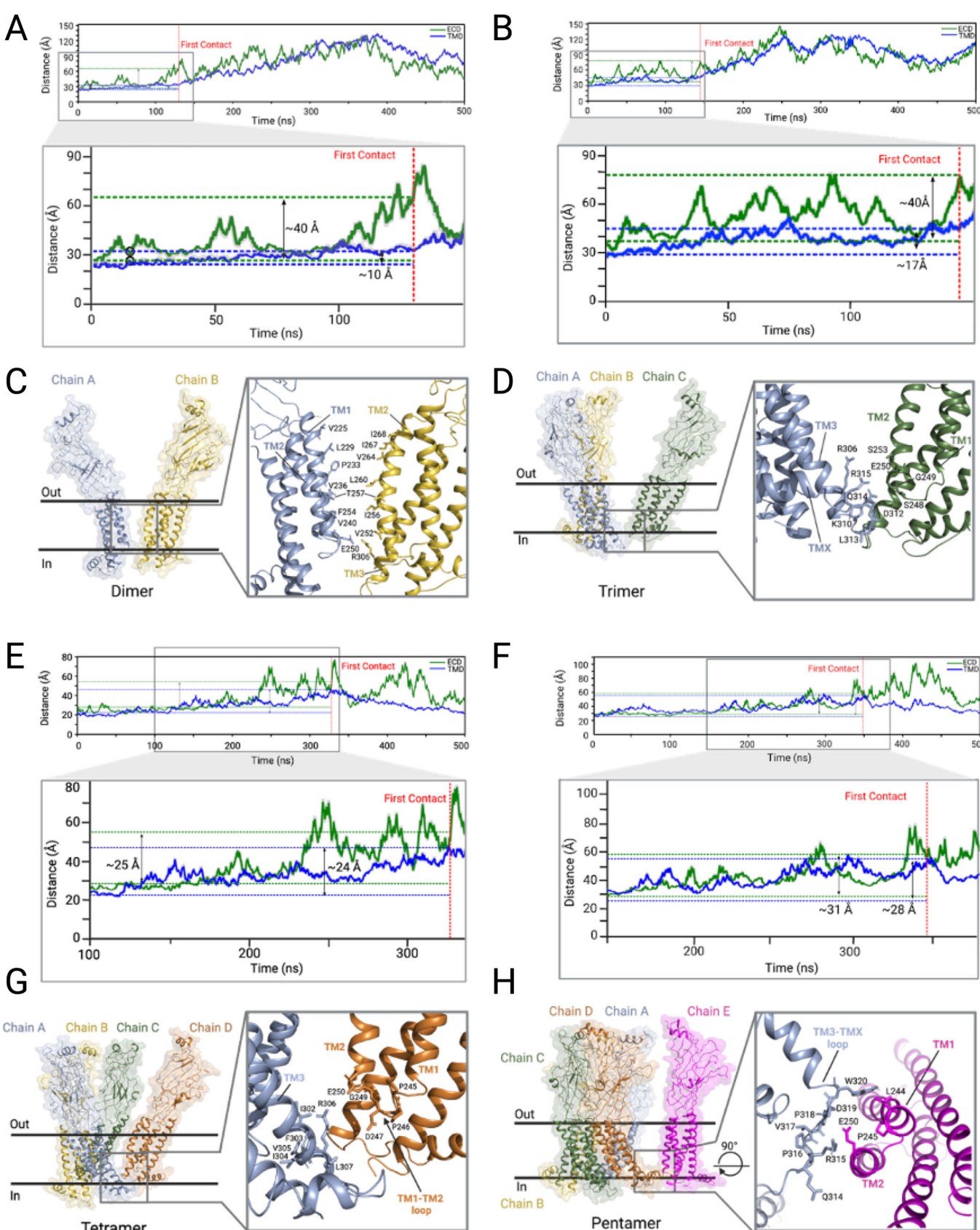

◄     **Figure 7.  Formation of 5-HT₃ₐR oligomers from precursors simulated by guided MD.**

Graphs depict the closest distance between ECDs (green) and TMDs (blue) centers of an oligomer over simulation time. Subunits were pulled apart; invert time scale for assembly view. Close-ups show TMD centroid distances at first contact, e.g., 10 Å (**A**), 17 Å (**B**), marked by red dashed lines. All curves are triplicate averages. Models display the first contact residues for each simulation: dimer (**A**, **B**), trimer (**C**, **D**), tetramer (**E**, **F**), and pentamer formation (**G**, **H**). Source data are available online for this figure.

influence receptor trafficking dynamics. Given the pivotal role of N-glycosylation in trafficking processes (Monk et al, 2004), the glycosylation of tetramers may serve as a facilitator for such trafficking events. Finally, understanding the sequence and mechanism of assembly presents opportunities for innovative pharmacological interventions targeting membrane protein assembly. The use of specific small molecules, peptides, or nanobodies that interact at the multimerization interface may modulate the amount of target protein. A similar approach has been suggested for blocking Aβ42 (Linse et al, 2022) polymerization and may be extrapolated to drug targets involving membrane proteins.

# Methods

## 5-HT₃ₐR expression, purification, and reconstitution in saposins

Murine wild-type 5-HT3A receptor containing four N-terminal StrepII tags was expressed by a stable T-Rex-293 cell line. As previously performed and described by us (Zhang et al, 2021), protein expression was induced by adding tetracycline to the cultures and by lowering the shaker's temperature to 30 °C. In order to increase protein expression levels (Chaudhary et al, 2012; Gorman et al, 1983), sodium butyrate was added shortly after the induction. The following steps were performed at 4 °C, unless differently specified. For the purification of the receptor, cells were resuspended in lysis buffer (10 mM HEPES, 1 mM EDTA, pH 7.4 plus 2 μg/mL leupeptin, 2 μg/mL pepstatin A, 0.2 mg/mL benzamidine-HCl, 20 μg/mL AEBSF, and 3 μg/mL aprotinin) using a dounce homogeniser and subsequently disrupted using a Microfluidizer processor (Microfluidics M-110L) at 80 psi. Cell debris was removed by centrifugation (10,000 × g, 45 min) and the membrane pellet was isolated from the clarified lysate by ultracentrifugation for 6 h at 130,000 × g. The membrane pellet was mechanically resuspended in solubilization buffer (final concentrations: 50 mM Tris, 500 mM NaCl, C12E9 (Anatrace) 0.75% (w/v), pH 8) and subsequently, it was left under gentle stirring for 2 h. Non-solubilised material was removed by centrifugation for 1 h at 49,000 × g. The supernatant was then passed through a 0.45-μm-pore filter and applied to a 5 mL Streptactin Superflow high-capacity column (IBA) (equilibration and washing buffer: 50 mM Tris, 150 mM NaCl, 0.01% C12E9, pH 8, flow rate of 1 mL/min). After the washing of the column with 250 mL of washing buffer, the receptor was eluted with the above-mentioned buffer supplemented with 10 mM D-desthiobiotin. The peak fractions were pooled and concentrated to ~300 μL for the further reconstitution of the channel into saposins and porcine brain polar lipids (BPL, Avanti Polar Lipids) (i.e., 5-HT3R-Salipro). Briefly, Once the lipids were prepared as previously described (Zhang et al, 2021), the purified channel was first incubated for 15 min at room temperature on an

orbital rocker with a 450-fold molar excess of detergent-destabilized BPL (650 g/mol). Secondly, saposin was added to 30-fold molar excess relative to 5-HT₃ₐR and the sample was incubated for 15 min at room temperature. Subsequently, the sample was incubated for 15 min at room temperature with 100–200 mg of wet BioBeads SM2 resin (Bio-Rad), and this last step was repeated in total two times. The sample was then filtered through a 0.22 μm-pore centrifugal filter and injected into a Superose 6 Increase 10/300 column (GE Healthcare) (SEC buffer 25 mM HEPES, 125 mM NaCl, pH 7.4). SEC peak fractions were pooled together and sample concentration was adjusted upon checking by negative staining using a Tecnai Spirit Biotwin at 120 kV (ThermoFisher Scientific).

## Unfolding of 5-HT₃ₐR in SDS

Blue native PAGE and circular dichroism (CD) measurements were performed as described previously (Tol et al, 2013). *Electrophoresis:* Pre-casted 5–20% gradient gels (Bio-Rad) were loaded with 10 μg of solubilized protein in sample buffer (final concentrations 10% glycerol, 0.2% Coomassie brilliant blue G-250 and 20 mM 6-aminohexanoic acid). NaCl concentration was kept below 25 mM, and the nonionic detergent concentration was not higher than 10 mM. Electrophoresis was performed at 4 °C, starting with 20 min at 25 V, until all samples entered the gel, followed by 2 h at 125 V, until the Coomassie front was leaking in the anode buffer. When the gel run was finished, excess Coomassie dye from empty detergent micelles was removed by destaining, and protein bands were accentuated by normal staining. The oligomers of BSA (ranging from 66 to 264 kD) were used as a molecular mass marker. CD measurements were performed as described before, on a model 62DS CD spectrometer (Aviv Biomedical, USA). Solubilized 5-HT3R was diluted to a concentration of 150 μg/ml in KPi buffer (10 mM potassium phosphate pH 7.4, with 0.025% (w/v) C12E9). In case the stock concentration was smaller than 2 mg/ml, the sample was dialyzed for several hours against the same buffer to prevent high UV-absorption buffer components. CD-spectra were recorded in 1 mm quartz cuvettes at a sampling interval of 1 nm, a bandwidth of 1.5 nm, and an average time of 5 s. Generally, spectra were measured from 260 to 190 nm. Spectra were smoothed by Savitzky-Golay filtering with 7 points and a second-order polynomial using Igor Pro (Wavemetrics) and converted to molar ellipticity values.

## Sample vitrification, cryo-EM data collection

Cryo-EM grids were prepared in the presence (i.e., holo) and absence (i.e., apo) of natural agonist serotonin (5-HT). In this case, we used photo-caged serotonin (RuBi-5-HT, Aabcam) that was activated by exposure to light. To observe the active form of the channel (i.e., holo active), the receptor was incubated in ice for 1 h

before freezing with 100 µM 5-HT and 2 mM CaCl₂. 3 µL (for apo and holo active samples) or 2 µL (for holo deactivated samples) of 5-HT3ₐR reconstituted in Salipro (Frauenfeld et al, 2016) at a final concentration of 0.01–0.07 mg/mL were applied onto freshly glow-discharged (15 mA for 45 s in a PELCO easiGlow system) Cu/C, R2/2, 300 or 400 mesh copper grids (Quantifoil) coated with a ~2 nm thick carbon layer. For the holo-deactivated sample, 1 µL of 600 µM caged serotonin (RuBi-5-HT, Abcam, final concentration on grid 200 µM) was directly added to 2 µL of the sample directly on the grid. Caged serotonin was used for further research reasons. For this experiment, the two components were incubated in dark conditions for 1 min on the grid and the ligand was further released upon controlled illumination with a harmless LED light source operating at 445 nm (blue) inserted in one of the lateral apertures of Vitrobot Mark IV (Thermo Fisher Scientific) right before the grid fell into liquid ethane.

All samples were blotted for 4.5–5.5 s with blot force 20 (595 Whatman paper) at 4 °C in 100% humidity and plunge-frozen into liquid ethane using a Vitrobot Mark IV (Thermo Fisher Scientific). Cryo-EM data were collected automatically using EPU software (Thermo Fisher Scientific) on a Titan Krios G3i microscope at 300 kV, equipped with a K3 (Gatan) detector operating in electron counting mode and a Bioquantum energy filter (Gatan). Movies were acquired at a nominal magnification of 105,000×, resulting in a pixel size of 0.837 Å. Total accumulated exposure of all the specimens ranged from 40 to 60 e⁻/Å². For both the apo and the holo-deactivated samples two data sets were collected from different preparations while for the holo-active sample, two data sets were collected on different dates from the same grid. Further data collection details are provided in Appendix Table S1.

## Isolation 5-HT3R vesicle and sample preparation and cryo-ET data collection

5-HT3R was expressed in the T-Rex-293 cell line as described earlier in Methods. After cells were collected, the supernatant of the growth medium was centrifuged at 3000 × g for 30 min to remove large vesicles and cell fragments that were out of interest. Microvesicles were collected from the pellet after supernatant centrifugation at 10,000 × g for 30 min. Microvesicles were resuspended in 500 µl PBS and hydrated overnight at 4 °C. Microvesicles were incubated with 2 mM CaCl₂ and 100 µM serotonin for 40 min on ice before vitrification. The concentration of microvesicles was checked via negative staining. Grids of Quantifoil Au/Au R2/2 were glow-discharged on the PELCO easiGlow system using 15 mA for 45 s. Four µl of microvesicle solution was applied to the grid and then vitrified by Vitrobot Mark IV (Thermo Fisher Scientific) adopting blot time of 4 s and blot force of 10.

Tilt series were recorded on the Thermo Fisher Scientific Titan Krios G3i in Core Facility for Cryo Electron Microscopy, Charité – Universitätsmedizin Berlin. Tilt series of +/−48° were acquired at a nominal magnification of 64,000× (0.69 Å per pixel in super-resolution mode) using SerialEM with a Hybrid-STA script (Sanchez et al, 2020) or PACEtomo (Eisenstein et al, 2023). The electron dose was 29.3 and 4.9 e⁻/Å² for non-tilted and tilted projections, respectively. Frame stacks were recorded on a Gatan K3 electron detector at the defocus range of −3 ~ −5 µm.

## Cryo-ET data processing and subtomogram averaging

Raw images output by electron microscope were directly imported to TomoBEAR (Balyschew et al, 2023), a workflow engine for streamlined processing of cryo-ET data for subtomogram averaging. The raw images were sorted according to different tilt series. MotionCor2 (Zheng et al, 2017) was used for correcting beam-induced motion, IMOD (Kremer et al, 1996) was used for generating tilt stacks, the fiducial beads were detected by Dynamo Tilt Series Alignment (Coray et al, 2024) and GCTF (Zhang, 2016) was used for defocus estimation. IMOD was then used for fiducial model refinement and tomographic reconstruction. The coordinates of particles were detected by template matching implemented in TomoBEAR. Candidate particles (30,623) were cropped and contaminations were excluded by Dynamo multi-reference classification (Castaño-Díez et al, 2012) adopting a local angular search in the pixel size of 11.04 Å. After several times of cleaning, classes of tetramer and pentamer were observed during classification in Dynamo when using a tight mask and searching for 360° for the in-plane Euler angle in the pixel size of 5.52 Å. Pentamer classes and tetramer classes were imported separately to the subtomogram averaging pipeline of Relion 4.0 (Zivanov et al, 2022). For pentamer and tetramer classes, duplicated particles (distance <50 Å) were removed. Relion 3D classifications were executed using local search to exclude contaminations and heterogeneity particles. Finally, 3D refinements were executed at the pixel size of 5.52 Å, and resolutions were estimated by Fourier shell correlation. The data flow is presented in Fig. EV3.

## Image processing for single particle cryo-EM

During data collection all data sets were pre-processed using CryoSPARC Live (Punjani, 2021) for frame alignment, contrast transfer function (CTF) estimation, particle picking, and extraction with a box size of 330 pixels (288 Å). Particularly, 2D classification "on-the-fly" helped to monitor data quality including particle orientation and oligomeric state of the channel. In CryoSPARC (Punjani et al, 2017), further 2D classification and 3D hetero-geneous refinement (without imposing symmetry) allowed to separate the tetrameric particles from the pentameric. After an initial 3D non-uniform refinement, tetramers' 3D refined particles together with a real-space mask excluding the Salipro belt and the solvent were used as inputs for 3D variability analysis (3DVA) (Punjani and Fleet, 2021). 3DVA was run using three variability components and a low-pass filter resolution of 6 Å. By using the *cluster mode* function in CryoSPARC in both the apo and holo deactivated data sets, particles were resolved into two main clusters that yielded both the symmetric and asymmetric maps of the tetramer. The obtained particle sets were again subjected to a non-uniform refinement in CryoSPARC, which resulted in maps of 3.5 Å resolution for the apo asymmetric tetramer (~385 K particles), of 3.4 Å resolution for the apo symmetric tetramer (~230 K particles), of 3.3 Å resolution for the holo deactivated asymmetric tetramer (~338 K particles) and of 3.4 Å resolution for the holo deactivated symmetric tetramer (~296 K particles). Local resolution estimation was performed in CryoSPARC. All resolutions were estimated according to the Fourier shell correlation (FSC) 0.143 cut-off criterion of two independently refined half maps (Rosenthal and Henderson, 2003).

## Calculation of minimal neighbor distance among 5-HT3Rs on microvesicles

Minimal neighbor distances of pentamers and tetramers were calculated separately. For pentamers or tetramers, coordinates were extracted from the final .star files from the Relion 3D refinement jobs and split according to tomogram indexes. Foreach tomogram, Euclidean distances between each particle and every other particle were calculated, and the lowest value was chosen. The distribution range of minimal neighbor distances (0 to 687 nm) was split into 100 intervals and the probabilities of each interval were calculated and plotted. The probabilities of distances larger than 400 nm were low and were truncated from the graphs for clarity.

## Model building and analysis

The initial model was built manually in UCSF Chimera (Pettersen et al, 2004) with single subunits from the published Salipro-5-HT3R (PDB accession code 6Y5A and 6Y59) for guidance. ISOLDE (Croll, 2018) was used for the initial model fitting. Densities corresponding to N-linked glycosylation adjacent to N82, N148, and N164 at the ECD were resolved but not included in the model (Fig. 1B,D, in red). COOT (Emsley et al, 2010) together with PHENIX (Liebschner et al, 2019) were used for the iterative refinement of the models. Model quality was assessed with Molprobity (Chen et al, 2010). Structural Figures were prepared using UCSF ChimeraX (Meng et al, 2023). The figures in this paper were made using BioRender.com.

## Protein interaction interface analysis

PDB files were processed with the PISA (Protein Interfaces, Surfaces, and Assemblies) program (Krissinel and Henrick, 2007) which calculates interface and different components of the binding free energy, including the solvation-free energy gain upon formation of the interface ($\Delta$iG, kcal/mol) and the associated $\Delta$iG P-value, which is a probability measure of the specificity of the interface.

## Protein structure preparations for simulations

All protein models were prepared with the Protein Preparation Wizard (Madhavi Sastry et al, 2013) included in Maestro under the OPLS4 force field (Lu et al, 2021): hydrogen atoms were added to the prepared cryo-EM structures at physiological pH (7.0) with the PROPKA tool (Bas et al, 2008) to optimize the hydrogen bond network. All water molecules were removed; C- and N-terminal cappings were added; disulfide bonds were assigned and constrained energy minimizations were carried out on the full-atomic models until the RMSD of the heavy atoms converged to 0.3 Å.

## Molecular dynamics simulation

We used three MD flavors customized for the particular problems. For the analysis of the transition between the tetrameric forms (Fig. 3A±C), we performed an enhanced rotation MD simulation (Kutzner et al, 2011). We performed all-atom MD simulations to analyze the atomic details for the pore permeation by water molecules (Fig. 3E). In this case, we took advantage of the relatively small size of the simulation system. To study the oligomerization processes, we performed the coarse-grained MD combined with sampling by metadynamics. We chose this combination of methods because the whole system is extremely large after we separate the molecules to a large distance.

Membrane systems were built using the membrane building tool CHARMM-GUI (Jo et al, 2008) and the OPM webserver (Lomize et al, 2012) was used to align the experimental structures in the lipid bilayer. To mimic the lipid composition of brain polar lipid, we used POPC, POPS, POPE, and cholesterol at a molar ratio of 15:22:39:24. Symmetrical and asymmetrical tetramer models were embedded in a lipid bilayer of 300 lipids, coupled to TIP3P water molecules comprising 0.15 M NaCl. The final systems contained 176,555 atoms and 177,295 atoms, respectively, with a volume of $100 \times 100 \times 180$ Å³. All systems were simulated using CHARMM36m force field (Huang et al, 2017). The MD simulations were performed in GROMACS (Berendsen et al, 1995). The systems were minimized with a 50,000-step energy minimization using the steepest descent algorithm. The systems were subjected to temperature equilibrating in the NVT ensemble at 310 K for 200 ps and then to density equilibrating in the NPT ensemble at 310 K and 1 atm for 10 ns. The heavy atoms were constrained using a harmonic restraint with a force constant set to 1000 kJ mol$^{-1}$ nm$^{-2}$ in the equilibrating steps. The production runs lasted 1 µs and three replicas were used to ensure reproducibility. The convergence of the simulation was confirmed by the analysis of RMSD of trajectories over time.

Since the subunit rotation is a rare biological event, the unbiased MD simulation might take an unexpectedly long time to sample. Therefore, for the simulations in Fig. 3A–C, we introduced a novel method to enforce the rotation changes between the symmetric and the asymmetric structure (Kutzner et al, 2011). It allows flexible adaptations of both the rotating subunit as well as the rotation axis during the simulation. Apart from the rotation itself, it imposes minimal constraints on the rotating group, allowing conformational adaptations to the surroundings. The radial profiles of proteins obtained from MD simulations were analyzed using CHAP (Klesse et al, 2019). The density of lipids was analyzed in the GROmaps tool. All Figures from MD simulations have been prepared in PyMOL (The PyMOL Molecular Graphics System, Version 1.2r3pre, Schrödinger, LLC). The parameters for MD simulations are shown in Appendix Tables S2–4.

## Metadynamics simulation

To probe how pentamers could form, we applied well-tempered metadynamics simulations, which is a reliable and widely used method for sampling rare biological events, such as protein-protein binding (Barducci et al, 2011, 2008). Metadynamics simulations were conducted using GROMACS-2021.5 (Berendsen et al, 1995) patched with plumed-2.7.3 (Bonomi et al, 2009). Coarse-grained (CG) simulation systems were built to investigate the assembly process of the 5-HT3R. We combined an Elastic Network with the Martini3force field (Souza et al, 2021) to obtain the initial atomic models. All CG systems were embedded in a phosphatidylcholine lipid bilayer (as described above) and were solvated in Martini CG water boxes, containing 0.15 M NaCl. In the CG simulations, a 5000-step energy minimization was then followed by a 5-ns NPT pre-equilibration simulation in which the positional constraints of the protein backbone beads were gradually relaxed. In the simulations, the temperature was set to 310 K using the V-rescale thermostat, and the pressure was set to 1 bar using a semi-isotropic coupling method. Free-energy profiles of the systems were calculated. Metadynamics simulations use a history-dependent potential $V(s, t)$ to accelerate the sampling of the collective

variables (CVs), $s$ ($s_1$, $s_2$, …, $s_m$). $V(s, t)$ is usually constructed as the sum of multiple Gaussians centered along the trajectory of the CVs:

$$V(s,t) = \sum_{j=1}^{n} G(s, t_j) = \sum_{j=1}^{n} w \prod_{i=1}^{m} exp\left(-\frac{[s_i(t) - s_i(t_j)]^2}{2\sigma^2}\right)$$

To achieve this, a Gaussian-shaped potential is added to bias the system at the current position of the CVs, periodically during the simulation. The height of the Gaussian is decreased with the amount of bias already deposited according to

$$w = w_0 e^{-\left[\frac{V(s,t)}{\Delta t}\right]} \tau_G$$

In our simulations, the distance between the mass center of the TMD and that of the ECD for each subunit was assigned as the CVs, while the width of Gaussians, $\sigma$, was set as 0.3. The time interval $\tau$ was 1.0 ps. Well-tempered metadynamics involves adjusting the height, $w_j$, in a manner that depended on $V(s, t)$, where the initial height of Gaussians $w_0$ was 1.5 kcal/mol, and the simulation temperature was 310 K Each metadynamics simulation lasted for 500 ns. The convergence of the simulations was confirmed by the analysis of distance plots over time. More details of metaMD simulations can be found in our previous works (Yuan et al, 2014; Chan et al, 2018).

## Data availability

The EM maps have been deposited to EMDB: EMD-16387 (5-HT-bound, asymmetric), EMD-16386 (5-HT-bound, symmetric), EMD-16384 (apo, asymmetric) and EMD-16385 (apo, symmetric) with the corresponding atomic maps deposited to PDB ID: 8C21 (5-HT-bound, asymmetric), 8C20 (5-HT-bound, symmetric), 8C1W (apo, asymmetric) and 8C1Z (apo, symmetric). Subtomogram averages were deposited to EMDB with the accession codes EMD-19419 (tetramer) and EMD-19420 (pentamer).

The source data of this paper are collected in the following database record: biostudies:S-SCDT-10_1038-S44318-024-00191-5.

## Peer review information

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

## Acknowledgements

The work was supported by the Sofja Kovalevskaja Award from the Alexander von Humboldt Foundation and the KU3222/2-1 KU3222/3-1 grants from the German Research Fund (DFG) to Mikhail Kudryashev. Mikhail Kudryashev is supported by the Helmholtz Association. Shuguang Yuan and Horst Vogel are supported by the Chinese Academy of Sciences and the Shenzhen Institute of Advanced Technology. Further funding from the following sources is acknowledged: Shenzhen government (grant no. JCYJ20200109114818703 to SY); Guangdong province (grant no. 2019QN01Y306 to SY); Shenzhen Key Laboratory for Computer Aided Drug Discovery at SIAT (ZDSYS20201230165400001 to SY and HV); the AlphaMol and SIATJoint Laboratory to SY and HV. The Guangdong Retired Expert, The Chinese Academy of Science President's International Fellowship Initiative (PIFI) (2020FSB0003), the NSFC-SNF grant (No. 32161133022, Shenzhen Pengcheng Scientist, the Shenzhen Government Top-Talent Working Funding and the Guangdong Province Academician Work Funding to HV. The authors thank Werner Kühlbrandt and Uljana Kravcenko for useful discussions. We thank the Central Electron Microscopy Facility of the Max Planck Institute of Biophysics and Dr. Sonja Welsch for support in the acquisition of the single particle cryo-EM data. We thank the Core Facility for cryo-Electron Microscopy (CFcryoEM) of the Charité - Universitätsmedizin Berlin for support in the acquisition of the data. CFcryoEM was supported by the DFG through grant No. INST 335/588-1 FUGG and the Berlin University Alliance (BUA). We thank Özkan Yildiz and Juan Castillo from the Max Planck Institute of Biophysics for the IT support at the Max Planck of Biophysics and the high-performance computing team at the MDC for supporting our operation at the Max-Cluster.

## Author contributions

**Bianca Introini**: Conceptualization; Data curation; Formal analysis; Validation; Investigation; Visualization; Writing—original draft; Writing—review and editing. **Wenqiang Cui**: Data curation; Software; Formal analysis; Validation; Investigation; Visualization; Methodology; Writing—original draft; Writing—review and editing. **Xiaofeng Chu**: Data curation; Formal analysis; Validation; Investigation; Visualization; Methodology; Writing—review and editing. **Yingyi Zhang**: Data curation; Formal analysis; Investigation; Methodology; Writing—review and editing. **Ana Catarina Alves**: Formal analysis; Validation; Investigation; Writing—review and editing. **Luise Eckhardt-Strelau**: Investigation. **Sabrina Golusik**: Investigation. **Menno Tol**: Formal analysis; Investigation. **Horst Vogel**: Supervision; Conceptualization; Formal analysis; Funding acquisition; Validation; Visualization; Methodology; Writing—original draft; Project administration; Writing—review and editing. **Shuguang Yuan**: Conceptualization; Data curation; Software; Formal analysis; Supervision; Funding acquisition; Validation; Investigation; Methodology; Writing—original draft; Project administration; Writing—review and editing. **Mikhail Kudryashev**: Conceptualization; Data curation; Software; Formal analysis; Supervision; Funding acquisition; Validation; Investigation; Methodology; Writing—original draft; Project administration; Writing—review and editing.

Source data underlying figure panels in this paper may have individual authorship assigned. Where available, figure panel/source data authorship is listed in the following database record: biostudies:S-SCDT-10_1038-S44318-024-00191-5.

## Funding

## Disclosure and competing interests statement

HV and SY are cofounders of AlphaMol Science Ltd.

# Expanded View Figures

## Only the extracellular domain reorganizes upon 5-HT binding

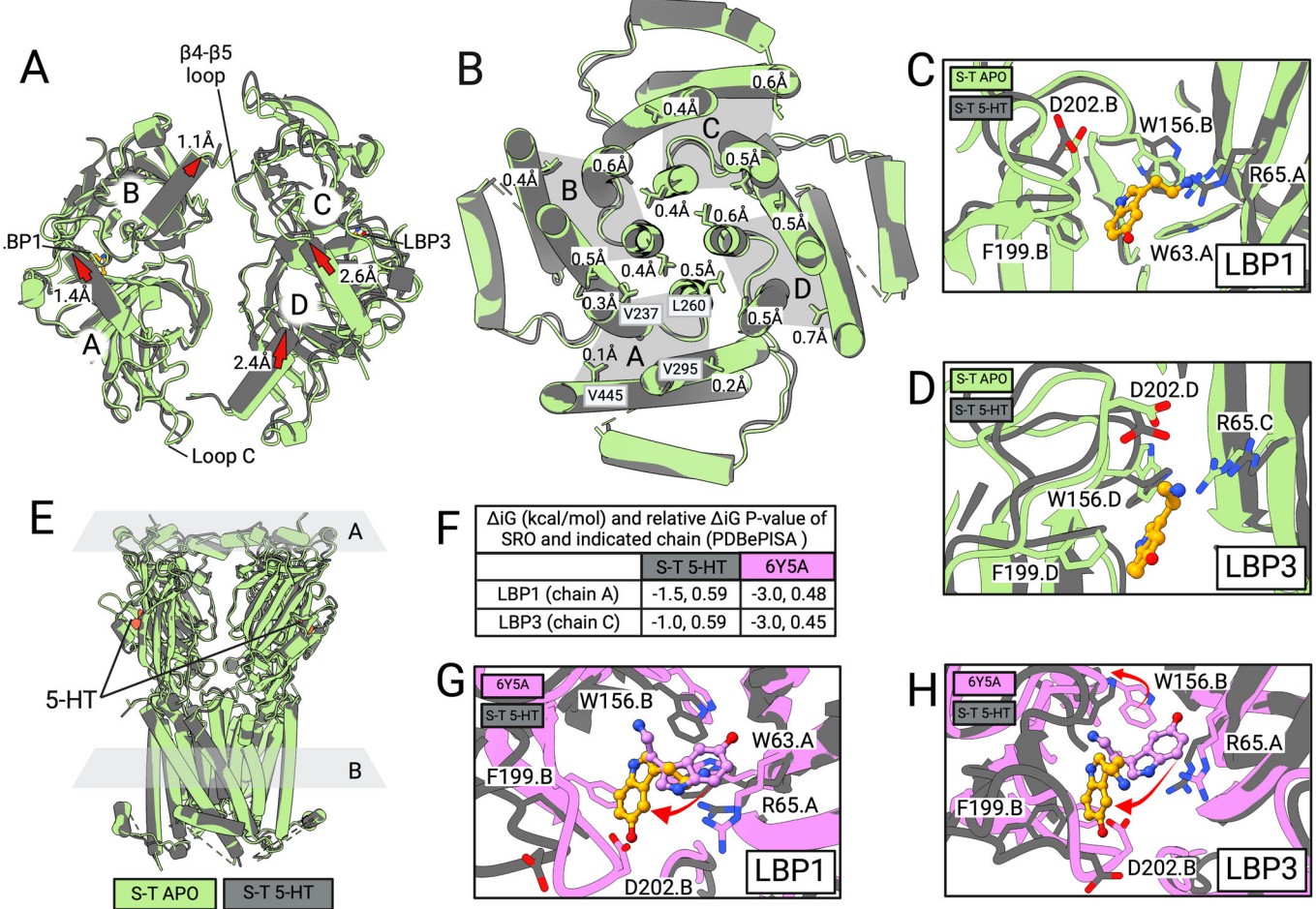

**Figure EV1.　The binding of serotonin does not lead to conformational changes in the transmembrane domain.**

(**A, B**) Superposition of apo- (i.e., S-T APO) and serotonin-bound (i.e., S-T 5-HT) structures for the symmetric tetramer, with serotonin densities in dark orange. Red arrows indicate N-terminal helices movement. Serotonin densities were resolved in two LBPs for the HOLO conformation. (**B**) Cross-sections at the TMD residue L260 (9' position) of M2 show no large movements. Displacements measured at the Cα atoms of indicated residues on each helix are shown. (**C, D**) Close-up of the LBPs of the apo and serotonin-bound tetramers, with serotonin in orange. (**E**) Depicts the planes shown in (**A**) and (**B**) and shows the locations of 5-HT molecules in red. (**F**) Solvation-free energy gain (ΔiG, kcal/mol) on the formation of the interface between indicated chains and ligands, calculated using PDBePISA server (Krissinel and Henrick, 2007). Negative values indicate a hydrophobic interface, with more negative values indicating stronger interaction. The ΔiG *P*-value is defined in (Krissinel, 2009). It is a measure of the specificity of the interface, *P*-values less than 0.5 suggest a specific molecular interaction. PDB IDs of each tetrameric form are reported above each panel. (**G, H**) Superpositions of the LBPs occupied by serotonin of the HOLO tetrameric 5-HT3_AR with the respective LBPs of the pentameric 5-HT5HT3_AR (PDB ID:6Y5A). Serotonin molecules bound to the tetramers are displayed in orange, and those bound to 6Y5A are in lavender.

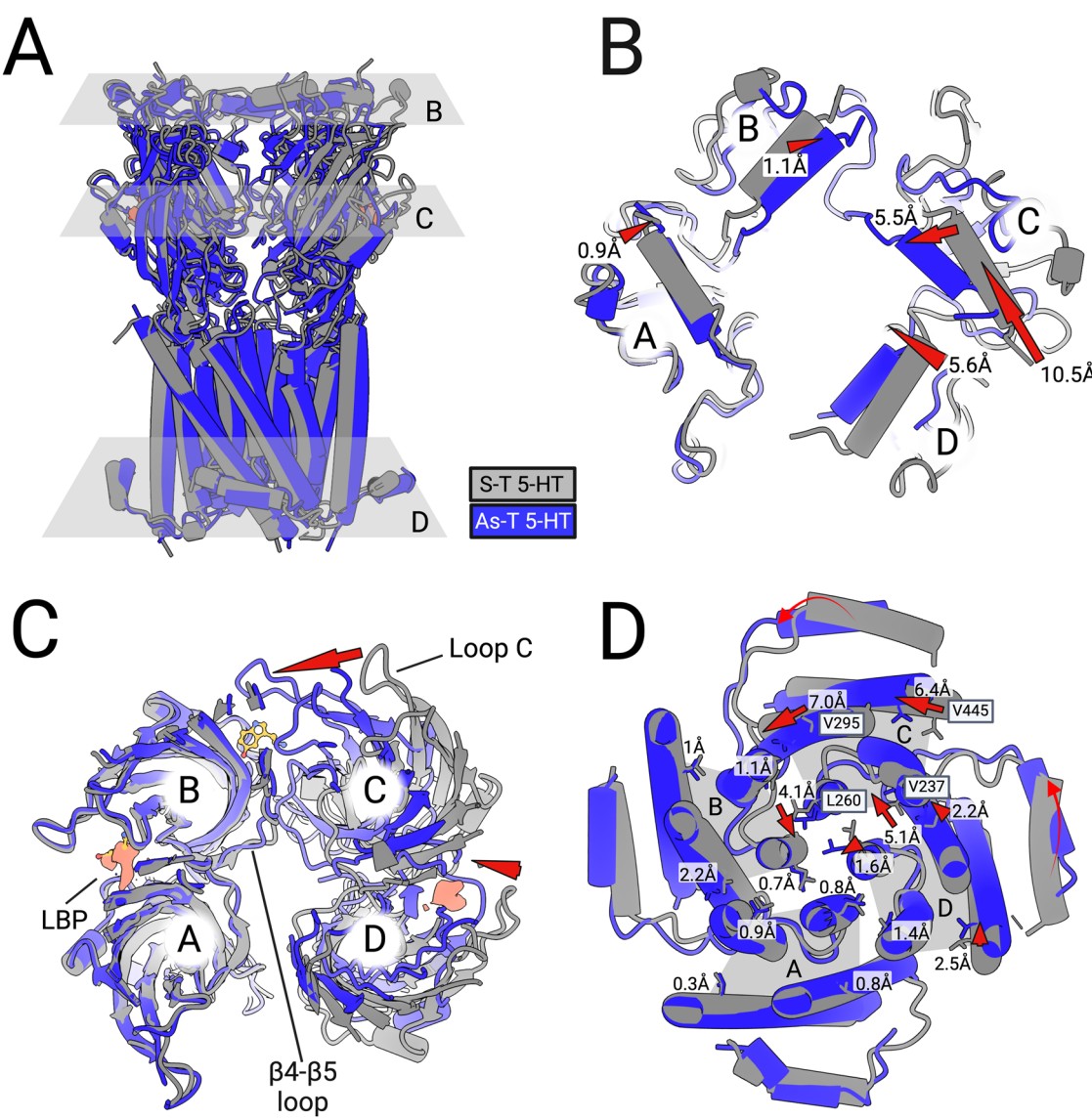

**Figure EV2.  The transition from symmetric to asymmetric tetramer involves the major movement of two subunits.**

Superposition of the models of the holo symmetric and holo asymmetric tetramers when aligned to subunit A. (A) Lateral view of the indicated superposed channels with the positions of the cross-sections analyzed in (B), (C), and (D). The models in (A) are visualized as chain traces. (B) Top view of the superposed channels. Red arrows indicate the direction of the movement of the N-terminal helices. (C) Cross-section at LBPs Serotonin molecules from the symmetric holo tetramer is shown in red, while those from the asymmetric holo tetramer are in orange. (D) Cross-sections at the TMD residue L260 (9' position) of M2. Displacements measured at the Cα atoms of the indicated residues on each helix in the same cross-section are shown. Also, in the presence of the ligand, both C and D helices are experiencing a downward movement while transitioning from symmetric to asymmetric.

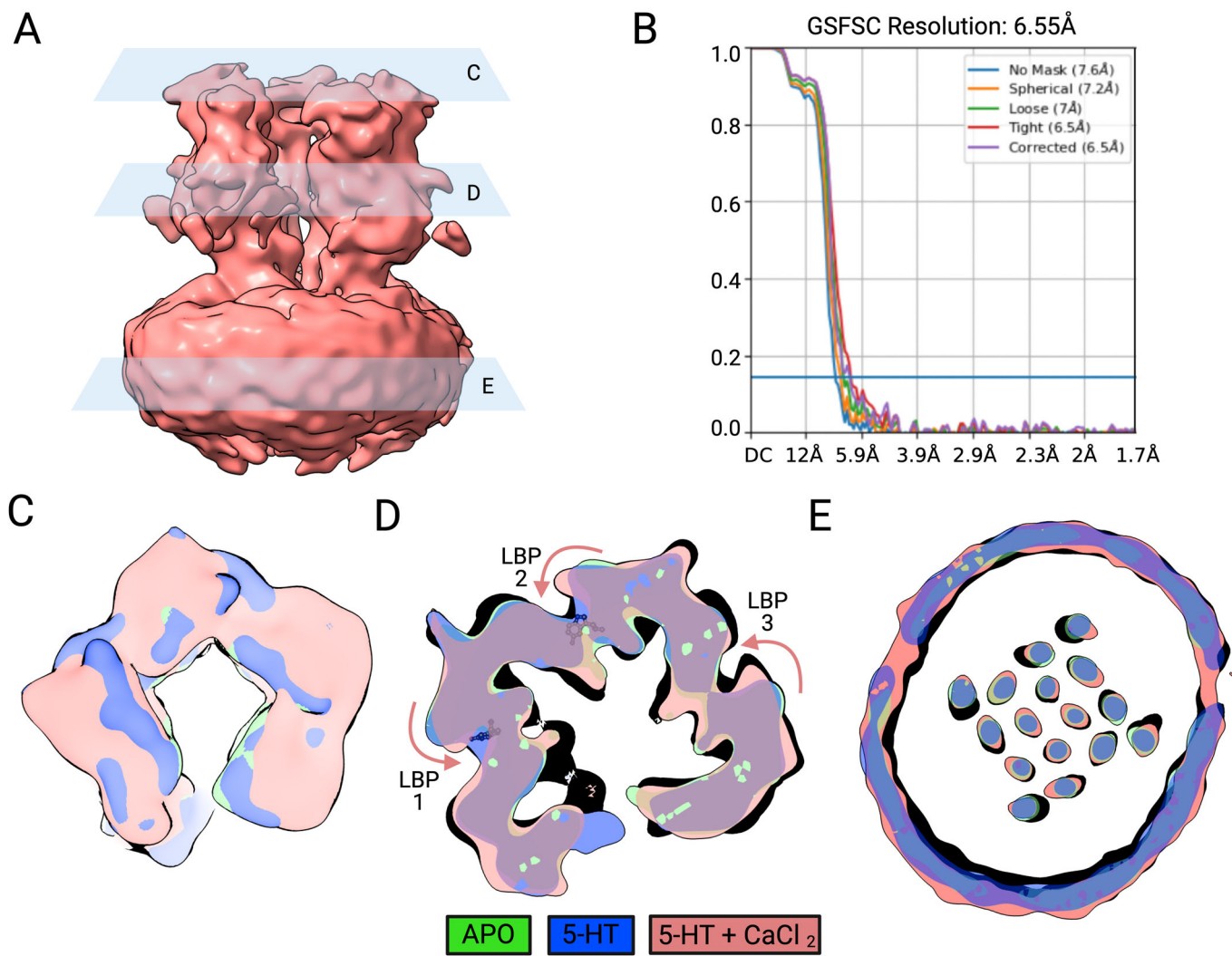

**Figure EV3. The addition of CaCl₂ and serotonin does not cause relevant movements at the transmembrane level.**

(A) Cryo-EM map of asymmetric 5-HT3AR tetramer in the presence of serotonin and 2 mM CaCl2 at a resolution of 6.55 Å. Positions of the cross-sections analyzed in (C), (D), and (E) are highlighted. (B) Estimating the resolution of the maps by FSC. (C–E) Maps of the asymmetric tetramer in the APO (green), HOLO desensitized (blue) and HOLO with CaCl₂ (salmon) were superposed and gaussian-filtered (standard dev 2.51) to facilitate comparison. (C) top view of the superposed maps. (D) Cross-section at the level of the ligand binding pockets (LBPs). The serotonin molecules visualized as balls and sticks are from the asymmetric holo tetramer (PDB ID: 8C21). The map suggests that in the presence of CaCl₂, there is a closure of the loop C of the LBP. (E) Cross-sections at the TMD at the level of the residue L260 (9' position) of M2.

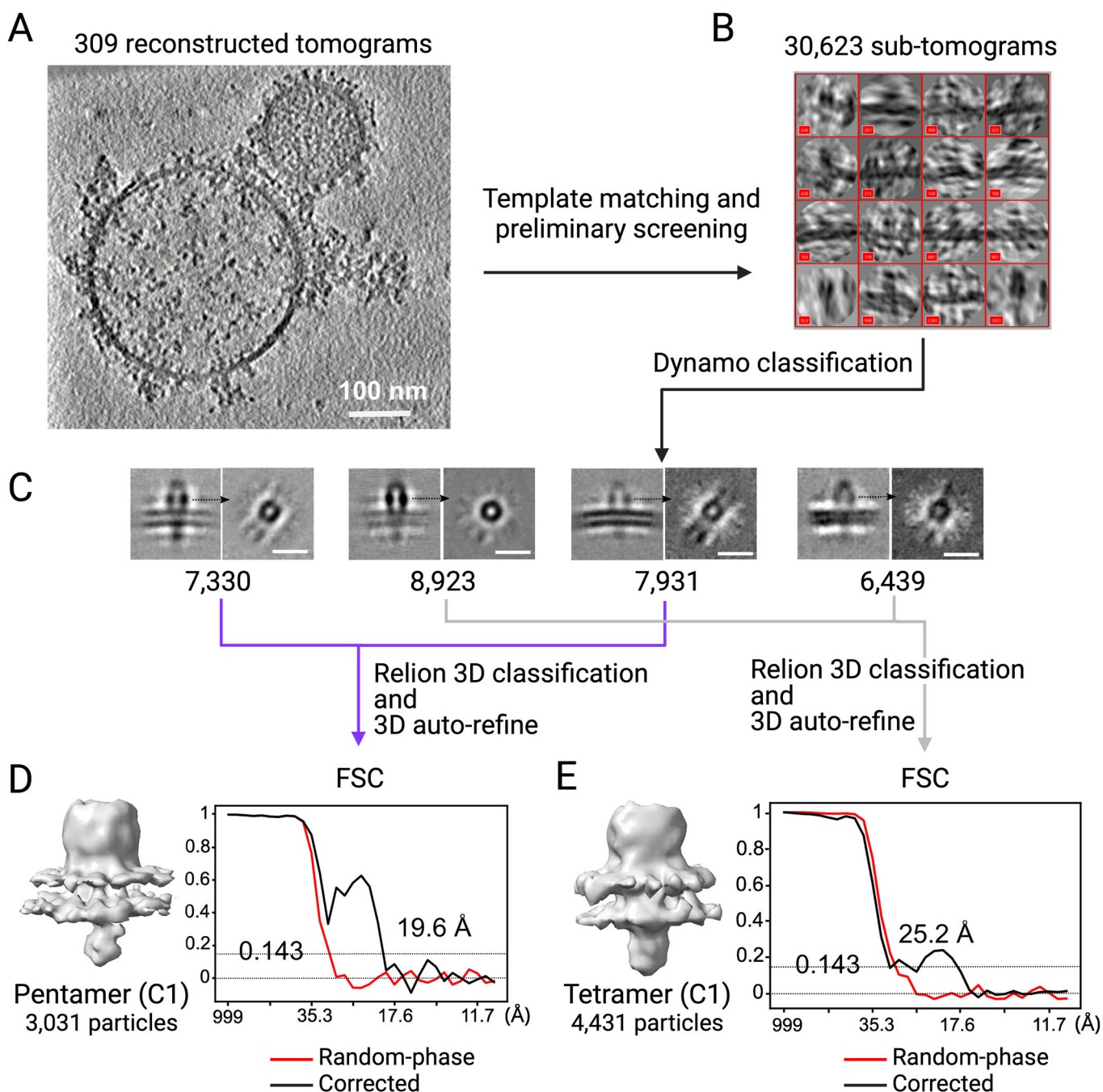

**Figure EV4. Overview of the workflow of cryo-ET and subtomogram classification and averaging.**

(A) Representative slice of a tomogram. (B) The particles were identified using GPU-accelerated template matching implemented in TomoBEAR (Balyschew et al, 2023). (C) Initially picked particles were subject to classification in Dynamo (Castaño-Díez et al, 2012). (D) Pentamer-looking classes were merged and processed to 19.6 Å resolution. and (E) the tetramer-looking classes were merged to produce a 25 Å map. No symmetry was applied.

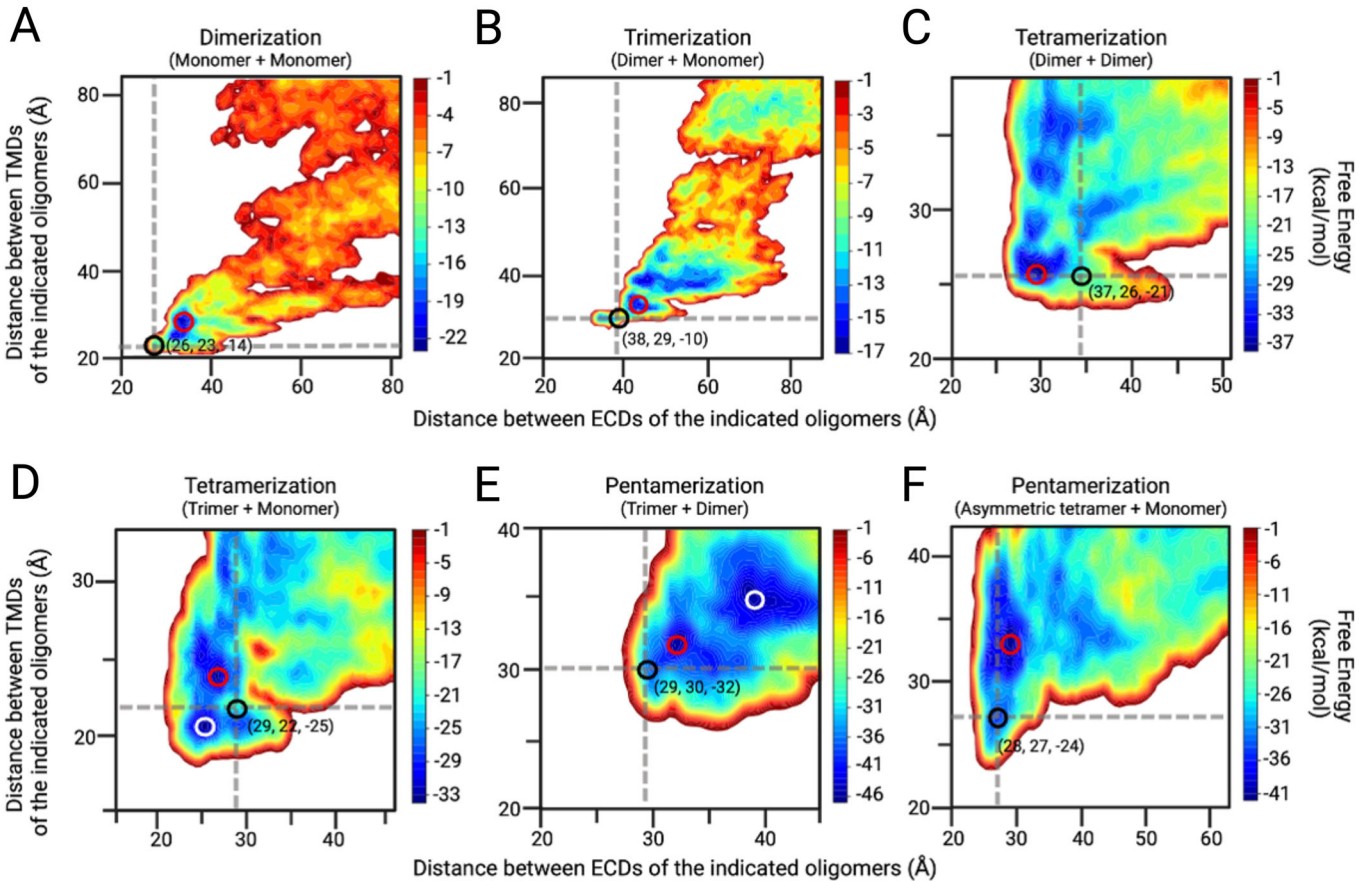

**Figure EV5. Free energy surfaces (FES) acquired from metadynamics simulations for the assembly process of the 5-HT3ₐR.**

(A–F) Free energy surfaces (FES) acquired from metadynamics simulations. X-axis represents the distance between the extracellular regions of the two subunits in Å, Y-axis represents the distance between the transmembrane regions of the two subunits in Å. The computed Gibbs free energy values (G) are expressed in kcal/mol and are represented by isoenergy lines drawn every 1 kJ/mol and shown in rainbow color. The black circle at the intersection of the two dotted lines represents the highest energy well (x, y, and z1) of the starting complex (i.e., the structure obtained from the cryo-EM experiment). The red circle represents the location of the lowest energy well (x, y, and z2) near the starting structure. A white circle represents a second low-energy well. The energy barrier is calculated as the difference between z1 and z2. The resulting value corresponds to the energy barrier (kcal/mol) that has to be overcome for the transition to a certain higher oligomeric organization. For each transition this value is depicted in Fig. 6G. A: monomer + monomer = dimer; B: dimer + monomer = trimer; C: dimer + dimer = symmetric tetramer; FD trimer + monomer = symmetric tetramer; E: trimer + dimer = pentamer; F: tetramer + monomer = pentamer.

