## [Peer Review File · The EMBO Journal]

Structure of Tetrameric Forms of the Serotonin-gated 5-HT_{3A} Receptor Ion Channel

Mikhail Kudryashev, Bianca Introini, Wenqiang Ciu, Xiaofeng Chu, Yingyi Zhang, Ana Catharina Alves, Luise Eckhardt-Strelau, Sabrina Golusik, Menno Tol, Horst Vogel, and Shuguang Yuan

Corresponding author(s): Mikhail Kudryashev (Mikhail.Kudryashev@mdc-berlin.de) , Shuguang Yuan (shuguang.yuan@cadd2drug.org)

Review Timeline:

Submission Date:	19th Mar 24
Editorial Decision:	3rd May 24
Revision Received:	28th May 24
Editorial Decision:	13th Jun 24
Revision Received:	16th Jun 24
Accepted:	15th Jul 24

Editor: William Teale

Transaction Report:

Dear Dr. Kudryashev,

Thank you again for the submission of your manuscript entitled "Structure of Tetrameric Forms of the Serotonin-gated 5-HT_{3A} Receptor Ion Channel" (EMBOJ-2024-117336) and for your patience during the review process. Your manuscript was sent for appraisal to two referees; we have now received the reports from both of them, which I copy below.

As you can see from their comments, while referee #2 indicated that the molecular dynamics simulations that you use need to be revisited and substantially improved, both point out the potential value of your work.

Based on the overall interest expressed in the reports, I would like to invite you to address the comments of all referees in a revised version of the manuscript. I should add that it is The EMBO Journal policy to allow only a single major round of revision and that it is therefore important to resolve the main concerns at this stage. I believe the concerns of the referees are reasonable and addressable, but please contact me if you have any questions, need further input on the referee comments or if you anticipate any problems in addressing any of their points. I am available to Zoom call and discuss the revisions at any time. Please, follow the instructions below when preparing your manuscript for resubmission.

I would also like to point out that as a matter of policy, competing manuscripts published during this period will not be taken into consideration in our assessment of the novelty presented by your study ("scooping" protection). We have extended this 'scooping protection policy' beyond the usual 3 month revision timeline to cover the period required for a full revision to address the essential experimental issues. Please contact me if you see a paper with related content published elsewhere to discuss the appropriate course of action.

Again, please contact me at any time during revision if you need any help or have further questions.

Thank you very much again for the opportunity to consider your work for publication. I look forward to your revision.

Best regards,

William

William Teale, Ph.D.
Editor
The EMBO Journal

When submitting your revised manuscript, please carefully review the instructions below and include the following items:

- 1) a .docx formatted version of the manuscript text (including legends for main figures, EV figures and tables). Please make sure that the changes are highlighted to be clearly visible.
- 2) individual production quality figure files as .eps, .tif, .jpg (one file per figure).
- 3) a .docx formatted letter INCLUDING the reviewers' reports and your detailed point-by-point response to their comments. As part of the EMBO Press transparent editorial process, the point-by-point response is part of the Review Process File (RPF), which will be published alongside your paper.
- 4) a complete author checklist, which you can download from our author guidelines ([https://wol-prod-cdn.literatumonline.com/pb-assets/embo-site/Author Checklist%20-%20EMBO%20J-1561436015657.xlsx](https://wol-prod-cdn.literatumonline.com/pb-assets/embo-site/Author%20Checklist%20-%20EMBO%20J-1561436015657.xlsx)). Please insert information in the checklist that is also reflected in the manuscript. The completed author checklist will also be part of the RPF.
- 5) Please note that all corresponding authors are required to supply an ORCID ID for their name upon submission of a revised manuscript.
- 6) We require a 'Data Availability' section after the Materials and Methods. Before submitting your revision, primary datasets produced in this study need to be deposited in an appropriate public database, and the accession numbers and database listed under 'Data Availability'. Please remember to provide a reviewer password if the datasets are not yet public (see <https://www.embopress.org/page/journal/14602075/authorguide#datadeposition>). If no data deposition in external databases is

needed for this paper, please then state in this section: This study includes no data deposited in external repositories. Note that the Data Availability Section is restricted to new primary data that are part of this study.

Note - All links should resolve to a page where the data can be accessed.

8) For data quantification: please specify the name of the statistical test used to generate error bars and P values, the number (n) of independent experiments (specify technical or biological replicates) underlying each data point and the test used to calculate p-values in each figure legend. The figure legends should contain a basic description of n, P and the test applied. Graphs must include a description of the bars and the error bars (s.d., s.e.m.).

9) We would also encourage you to include the source data for figure panels that show essential data. Numerical data can be provided as individual .xls or .csv files (including a tab describing the data). For 'blots' or microscopy, uncropped images should be submitted (using a zip archive or a single pdf per main figure if multiple images need to be supplied for one panel). Additional information on source data and instruction on how to label the files are available at .

10) We replaced Supplementary Information with Expanded View (EV) Figures and Tables that are collapsible/expandable online (see examples in <https://www.embopress.org/doi/10.15252/embj.201695874>). A maximum of 5 EV Figures can be typeset. EV Figures should be cited as 'Figure EV1, Figure EV2" etc. in the text and their respective legends should be included in the main text after the legends of regular figures.

12) Our journal encourages inclusion of *data citations in the reference list* to directly cite datasets that were re-used and obtained from public databases. Data citations in the article text are distinct from normal bibliographical citations and should directly link to the database records from which the data can be accessed. In the main text, data citations are formatted as follows: "Data ref: Smith et al, 2001" or "Data ref: NCBI Sequence Read Archive PRJNA342805, 2017". In the Reference list, data citations must be labeled with "[DATASET]". A data reference must provide the database name, accession number/identifiers and a resolvable link to the landing page from which the data can be accessed at the end of the reference. Further instructions are available at .

Further instructions for preparing your revised manuscript:

At EMBO Press we ask authors to provide source data for the main manuscript figures. Our source data coordinator will contact you to discuss which figure panels we would need source data for and will also provide you with helpful tips on how to upload

and organize the files.

We realize that it is difficult to revise to a specific deadline. In the interest of protecting the conceptual advance provided by the work, we recommend a revision within 3 months (1st Aug 2024). Please discuss the revision progress ahead of this time with the editor if you require more time to complete the revisions. Use the link below to submit your revision:

Referee #1:

The serotonin-gated 5HT3 receptor is a pentameric ligand-gated channel that has been well studied with a number of structures already published. The interest in this manuscript is the first structure determination of the 5HT3R in a tetrameric form, which formed a subset of the particles purified from heterologous expression in HEK293 cells. The biological role of a tetrameric 5HT3R is unknown as the central pore was found in MD simulations to be devoid of water molecules as it was considerably narrower than the channel found in pentameric channels. Two forms of tetramer were observed, symmetric and asymmetric and a series of theoretical calculations and MD simulations suggested that it is energetically plausible for these two forms to interconvert. In the symmetric tetramer, densities were observed in the ligand binding pocket consistent with the presence of serotonin and calculations suggested that one of the serotonin molecules in the asymmetric tetramer probably bound with a similar affinity to the pentameric channel, but the other showed less favourable binding, as did both serotonin molecules in the symmetric tetramer. Cryo-ET was then used to demonstrate the presence of the 5HT3R on the cell surface of HEK293 cells, suggesting that the cell regarded the tetramer as being properly folded (misfolded membrane proteins remain in the ER and are degraded) and thus could be of biological relevance. One of the most remarkable findings was that atomistic MD simulations suggested that the asymmetric tetramer could be an intermediate in the folding pathway, with a monomer being able to insert in the tetramer to form a pentamer.

Minor points

1. In the Discussion (first paragraph) there needs to be a discussion about the effect that the heterologous expression system used has on the production of the 5HT3R tetramers. The authors used a tetracycline inducible system in HEK293 cells. I agree that this is a human neuronal cell line, but the authors induce high levels of expression with tetracycline and sodium butyrate, which could alter the balance between tetramers and pentamers, resulting in far more tetramers than may be found in the brain.
2. There is no clear discussion on why serotonin binding to the tetramer does not result in channel opening. Given that the molecular mechanism for serotonin-induced channel gating is well known, it would be helpful to have a simplified picture (may be a cartoon) highlighting the key elements in the process and which elements cannot happen in the tetramer.
3. It would be helpful if there was a Table describing all the MD simulations and the relevant parameters for them. In addition, plots of the RMSD of trajectories over time should be presented to show that the simulations have equilibrated.
4. Line 117: It is stated that 'both conformations' have C4 symmetry, but how can an asymmetric channel have symmetry?
5. In Figure 1F, I am uncertain which bands represent pentamers and which represent tetramers as the gel has not been run long enough. There appears to be a step change between 0 and 0.11 lanes, but also between 0.33 and 0.45. Can this be improved?
6. In Figure 2A The colour scale should be the same in each chain and it should be reduced so that everything does not look the same colour. Using 0.05 to about 3-4 with rainbow coloration would be a good improvement.

Referee #2:

In this contribution by Introini et al., the EM structures of tetrameric form of 5-HT3A receptor are reported, which were further characterised by MD simulations to decipher the most plausible way of pentamer formation. The manuscript is in principle well-written (apart from Materials and Methods - that part is quite sloppy with many typos) and easy to follow.

The main strength is a robust structural confirmation of a tetramer existence, though the discussion should be somewhat extended - as this phenomenon is being increasingly reported these days (see for example DOI: 10.1038/s41586-023-06470-1, where pentamer of tetrameric TRPV3 is reported). Hence this study is very timely and should be of interest for the general audience.

The main weakness of the manuscript is that there is no real functional characterisation of the tetrameric form - the structural part is very solid (see a few minor comments below), but the functional part is in silico only, hence error-prone (given the fact that the structures are resolved at the moderate resolution). Therefore, as the least, the statements such as 'tetramers... bind ligands less efficiently' should be toned down.

The MD part of the manuscript is less straightforward. Some simulations were done in coarse-grain mode and some in atomistic mode, if I am not mistaken. What was the rationale for the atomistic calculations (and given the size of a system it is a bit of a concern). Also, what is about the time scale? In Fig. 3, some of simulations are only 1000picoseconds, and some are 1 μ s, and 1000ps (0.001 μ s) is clearly not enough. For the calculations of the assembly landscape, given the comparable energy barriers for tetramer formation via dimer and dimer or via trimer+monomer, the probabilities of both routes should be calculated, e.g. via Markov state modelling. Also could the authors perhaps explain why they went with meta dynamics and not umbrella sampling? Also the interconversion barrier of 0.67kcal is below kT at room temperature, so this should be a spontaneous process.

Minor concerns: did the authors try to do a symmetry relaxation for their single particle data? It can help getting better reconstructions. Also in table EV1 - no correlation coefficients reported, please add.

In Fig.2 panels D-G - please reconsider the colour scheme - hard to distinguish.

Please brush up the Materials and Methods section: there are no spaces between words and references, various spelling of words (e.g. min and minutes), some words are unnecessarily in capitals (e.g. Briefly, Once...), misspellings (e.g. Gantan detector, repaired cryo-EM structures), etc.

We thank the reviewers for their comments and suggestions. Our response is below in bold. We marked the related changes in the manuscript file “Tetramer_MS_Revision_tracked” with yellow and provided a “clean” version.

Referee #1:

The serotonin-gated 5HT3 receptor is a pentameric ligand-gated channel that has been well studied with a number of structures already published. The interest in this manuscript is the first structure determination of the 5HT3R in a tetrameric form, which formed a subset of the particles purified from heterologous expression in HEK293 cells. The biological role of a tetrameric 5HT3R is unknown as the central pore was found in MD simulations to be devoid of water molecules as it was considerably narrower than the channel found in pentameric channels. Two forms of tetramer were observed, symmetric and asymmetric and a series of theoretical calculations and MD simulations suggested that it is energetically plausible for these two forms to interconvert. In the symmetric tetramer, densities were observed in the ligand binding pocket consistent with the presence of serotonin and calculations suggested that one of the serotonin molecules in the asymmetric tetramer probably bound with a similar affinity to the pentameric channel, but the other showed less favourable binding, as did both serotonin molecules in the symmetric tetramer. Cryo-ET was then used to demonstrate the presence of the 5HT3R on the cell surface of HEK293 cells, suggesting that the cell regarded the tetramer as being properly folded (misfolded membrane proteins remain in the ER and are degraded) and thus could be of biological relevance. One of the most remarkable findings was that atomistic MD simulations suggested that the asymmetric tetramer could be an intermediate in the folding pathway, with a monomer being able to insert in the tetramer to form a pentamer.

Minor points

1. In the Discussion (first paragraph) there needs to be a discussion about the effect that the heterologous expression system used has on the production of the 5HT3R tetramers. The authors used a tetracycline inducible system in HEK293 cells. I agree that this is a human neuronal cell line, but the authors induce high levels of expression with tetracycline and sodium butyrate, which could alter the balance between tetramers and pentamers, resulting in far more tetramers than may be found in the brain.

Thank you, we added this important discussion point to the first paragraph of the discussion.

It reads: In our current work, as well as in the report by López-Sánchez and colleagues, the protein was produced in the tetracycline-inducible expression system in HEK293 cells, optimized for high protein yield, resulting in up to 107 receptors per cell (Hassaine et al, 2013). This is higher than the number of 5-HT3R per neuronal cell, therefore the ratio of the tetramers to pentamers in vivo might differ.

2. There is no clear discussion on why serotonin binding to the tetramer does not result in channel opening. Given that the molecular mechanism for serotonin-induced channel gating is well known, it would be helpful to have a simplified picture (may be a cartoon) highlighting the key elements in the process and which elements cannot happen in the tetramer.

Thank you very much, this is a very interesting point. Activation of the receptor is a tightly regulated process. It was shown for 5-HT3R that at least two ligands need to bind to non-consecutive ligand binding pockets for intermediate to long duration currents (Raves et al, Molecular Pharmacology, 2005, 68 (5) 1475-1483). As we pointed out in our analysis of the LBPs in tetramers, only one of the LBPs in the asymmetric tetramer seems to have a functionally bound ligand. This allows us to hypothesize that a functional LBP requires both monomers to have neighboring subunits.

We updated the second paragraph of the discussion to highlight this point. We also swapped Figure 4 and the former Figure EV4, now EV1 as it shows LBPs for the asymmetric tetramer, which seems more important in this context. More mechanistic insights could be gained by a long detailed MD simulation of tetramers with ligands, however, we believe that this goes beyond the scope of the current manuscript.

3. It would be helpful if there was a Table describing all the MD simulations and the relevant parameters for them. In addition, plots of the RMSD of trajectories over time should be presented to show that the simulations have equilibrated.

Thank you very much, indeed, all simulations in this work were equilibrated before MD production runs, which we mention in the methods section of the updated MS. The in-depth discussion of the equilibration process seems to be too basic, given the size limitations. In the updated version of the MS, we provide tables summarizing the details of the performed simulations (new tables EV2-EV4).

(1) Enforced rotation MD

This is a special advanced sampling simulation. It doesn't have a meaningful traditional RMSD over time.

Table EV2. Exemplary GROMACS .mdp file entries for enforced rotation MD

Parameters	ChainA	ChainB	ChainC	ChainD
rotation	Yes	Yes	Yes	Yes
rot-nstrout	1	1	1	1
rot-nstsout	10	10	10	10
rot-ngroups	4	4	4	4
rot-type0	iso	iso	iso	iso
rot-massw0	no	no	no	no

rot-vec0	-0.07 0.49 -0.87	-0.19 0.16 -0.97	0.38 0.29 0.88	0.55 0.12 0.82
rot-pivot0	7.05520 4.52350 10.27665	6.55805 3.99875 10.10755	4.15225 6.12650 9.88275	5.33165 6.69430 9.96785
rot-rate0	0.0046	0.00531	0.01891	0.00823
rot-k0	500	500	500	500
rot-fit-method0	norm	norm	norm	norm

(2) Conventional MD

Table EV3. Time and constraint potential parameters of each step

Steps	Types	dt/fs	Time/ ps	Restraint/KJmol-1 nm-2				
				Backbone	Sidechain	Ligand	Lipid	Dihedral
1	EM	1	50	4000	2000	4000	1000	1000
2	NVT	1	125	4000	2000	4000	1000	1000
3	NPT	1	125	2000	1000	2000	400	400
4	NPT	1	125	1000	500	1000	400	200
5	NPT	2	500	500	200	500	200	200
6	NPT	2	500	200	50	200	40	100
7	NPT	2	500	50	0	50	0	0
8	cMD	2	1×10 ⁶	0	0	0	0	0

(3) Metadynamics MD

This is another advanced sampling simulation. It doesn't have a RMSD. It sampled the energy barrier via different states.

Table EV4. Distance between the center of mass between ECD and TMD atoms

Types	CV1		CV2	
	COM1	COM2	COM1	COM2
112	1-536	914-1449	537-913	1450-1826
123	1-536, 914-1449	1828-2363	537-913, 1450-1826	2364-2743
134	1-536, 914-1449, 1827-2362	2738-3263	537-913 1450-1826 2363-2737	3264-3643
224	1-536, 914-1449	1827-2362, 2738-3263	537-913 1450-1826	2363-2737, 3264-3643

145	1-536, 914-1449, 1827-2362, 2740-3275	3653-4188	537-913, 1450-1826, 2363-2739, 3276-3652	4189-4565
235	1-536, 914-1449, 1827-2362	2740-3275, 3653-4188	537-913, 1450-1826, 2363-2739	3276-3652, 4189-4565

In the conventional MD simulations (eg. Figure 3E), we made the plots of the RMSD of trajectories over time to show that the simulations have equilibrated (as follows, the figure for the review).

We added a note to the methods section about MD simulations

For the other simulations, we use the distance plot instead of the RMSD plot because our goal is to measure the distance between different subunits (eg. Figure 7), rather than to assess the stability of the system. We also made a note about the convergence of simulations in the methods section.

4. Line 117: It is stated that 'both conformations' have C4 symmetry, but how can an asymmetric channel have symmetry?

Thank you, we meant that while the ECD was not symmetric, the transmembrane helices followed a C4 symmetry. We further clarified this in the text.

it reads: “Further classification of the tetrameric particles, without the application of symmetry, resulted in two structures: a near-C2-symmetric (which we will refer to as symmetric) and an asymmetric tetramer. The differences in the conformations were observed mostly in the extracellular domain (ECD): in the asymmetric tetramer, the ECD is organized similarly to the pentameric form of the receptor with one missing subunit (Fig 1B), while the second conformation assembles as an apparent dimer of dimers (Fig 1D). Both conformations showed tightly packed transmembrane domains (TMD) with a C4 symmetry between the membrane leaflets.”

5. In Figure 1F, I am uncertain which bands represent pentamers and which represent tetramers as the gel has not been run long enough. There appears to be a step change between 0 and 0.11 lanes, but also between 0.33 and 0.45. Can this be improved?

We do agree that in the native gel of Figure 1F the pentamers and tetramers are not very well resolved although lanes 0.82 and 0.95 indicate faintly the simultaneous presence of tetramers and pentamers. We assume that the step changes appearing between lanes 0 and 0.11, as well as between lanes 0.33 and 0.44 are induced by the presence of increasing SDS which changes the running properties of the 5-HT3 receptor oligomers as compared to the 0 lane. We observe the actual pentamers and tetramers in our cryo-EM data. The major goal of this native gel is to document the dissociation of the 5-HT3 receptor into lower oligomers which according to the CD spectra are not accompanied by conformational changes. As even better resolution between pentameric and tetrameric receptors in the native gel would not change our line of argumentation, we do not feel a requirement to exchange the figure at the present state of the manuscript.

We updated the text to point out that this experiment focuses on the lower oligomeric forms, it reads: “While our cryo-EM data demonstrates that the tetramers maintain the overall conformation of monomers, the unfolding experiments suggest that even lower-order oligomers maintain a stable conformation even in aggressive environments (Fig 1G).”

6. In Figure 2A The colour scale should be the same in each chain and it should be reduced so that everything does not look the same colour. Using 0.05 to about 3-4 with rainbow coloration would be a good improvement.

Thank you very much, we updated the color scheme in most of the figures, including Fig 2A.

Referee #2:

In this contribution by Introini et al., the EM structures of tetrameric form of 5-HT_{3A} receptor are reported, which were further characterised by MD simulations to decipher the most plausible way of pentamer formation. The manuscript is in principle well-written (apart from Materials and Methods - that part is quite sloppy with many typos) and easy to follow.

The main strength is a robust structural confirmation of a tetramer existence, though the discussion should be somewhat extended - as this phenomenon is being increasingly reported these days (see for example DOI: 10.1038/s41586-023-06470-1, where pentamer of tetrameric TRPV3 is reported). Hence this study is very timely and should be of interest for the general audience.

Thank you very much for the summary of our manuscript and your suggestions.

The main weakness of the manuscript is that there is no real functional characterisation of the tetrameric form - the structural part is very solid (see a few minor comments below), but the functional part is in silico only, hence error-prone (given the fact that the structures are resolved at the moderate resolution). Therefore, as the least, the statements such as 'tetramers... bind ligands less efficiently' should be toned down.

Thank you, we updated the section header to “Tetramers of the 5-HT_{3A}R are capable of binding serotonin” and highlighted that the functional suggestions are based on the poses of the ligand in the LBPs.

The corresponding section of the discussion reads: “Despite the high structural similarity between the individual monomers in the tetrameric and pentameric forms of the receptor, the inter-protomer interactions, as well as the poses of serotonin in the ligand-binding pockets are different. Interestingly, the only ligand binding pocket that showed the pose of the serotonin molecule similar to in a functional pentamer was the LBP2 of the asymmetric tetramer. LBP2 is located at the sole interface wherefor which each of the ECD domains also has neighboring domains that seemingly apparently stabilize the LBP, facilitating the efficient binding of serotonin.”

We also toned down a part of the last paragraph of the discussion: “Tetrameric forms of 5-HT_{3A}R, which may represent assembly intermediates preceding pentamers, are delivered to the plasma membrane, suggesting possible hinting at potential functional significance.”

The MD part of the manuscript is less straightforward. Some simulations were done in coarse-grain mode and some in atomistic mode, if I am not mistaken. What was the rationale for the atomistic calculations (and given the size of a system it is a bit of a concern).

Thank you very much, that is correct. Different MD simulations were performed for different purposes.

1. For quantifying the transition between the tetrameric forms (Figure 3 A-C), we performed enhanced rotation MD simulations. This is a previously reported customized MD simulation, which we describe in the methods section
2. For the pore permeation by water molecules (Figure 3 E), we performed all-atom MD simulations to analyze the atomic details. In this case, we took advantage of the relatively small size of the simulation system.
3. We performed the coarse-grained MD to study the oligomerization processes. We chose this method because the whole system is extremely large after we separate the molecules to a large distance. It is infeasible to perform all-atom MD for such a system.

We updated the methods section for the MD simulations (first paragraph) to outline these considerations.

Also, what is about the time scale? In Fig. 3, some of simulations are only 1000picoseconds, and some are 1 μ s, and 1000ps (0.001 μ s) is clearly not enough.

Thank you, the 1000 ps simulation refers to a special sampling of MD simulations, which is known as Enforced rotation MD described in the reference of our MS: Kutzner, Gzub and Grabmuller, 2011. This is different from the other traditional MD simulations. It carried out molecular dynamics simulations of the whole connector both in equilibrium and under mechanical stress. The simulations revealed a quite heterogeneous distribution of stiff and soft regions, resembling that of typical composite materials that are also optimized to resist mechanical stress.

This method enforces the rotation of a protein subunit in molecular dynamics (MD) simulations. This “flexible axis” approach allows flexible adaptations of both the rotating subunit as well as the rotation axis during the simulation. The rotary motion of subunits is an essential part of the function of many proteins. Such processes are typically too slow and/or too rare to be observed on the molecular dynamics (MD) time scales. However, rotation can be induced and/or increased in rate with the help of external forces that are exerted on certain subunits. The MD simulations for Figure 3A-C is the enhanced rotation MD. In this case, the time scale of 1000 ps allows adequate sampling.

To enforce rotation, a group of N atoms (the rotation group) is subjected to a rotation potential V. Each atom with position x_i gets assigned a reference (equilibrium) position $y_i(t)$, which is rotating at a constant angular rate ω about an axis v . Typically the initial positions of the rotation group provide the initial ($t=0$) set of reference positions y_i^0 . The

$$V^{\text{iso}} = \frac{k}{2} \sum_{i=1}^N w_i [\Omega(t)(y_i^0 - u) - (x_i - u)]^2$$

isotropic potential

with the rotation matrix Ω and the pivot u of the axis yields forces towards the (rotating) reference positions $y_i(t)$, thus effectively rotating x_i . With the prefactors

$$w_i = Nm_i/M$$

with total mass:

$$M = \sum_{i=1}^N m_i$$

mass-weighting can be achieved.

(A) Fixed vs. flexible axis rotation, (B) fixed axis, (C) flexible axis rotation.

We highlighted that we used the flexible axis approach in the methods section and in the main text (section on Transition between the tetrameric forms of 5-HT_{3A}R)

For the calculations of the assembly landscape, given the comparable energy barriers for tetramer formation via dimer and dimer or via trimer+monomer, the probabilities of both routes should be calculated, e.g. via Markov state modelling.

Thank you very much for the suggestion. However, we believe that our methods are sufficient for the given question: our current approach already considers the main energy barriers for tetramer formation through the dimer-dimer and trimer-monomer pathways, demonstrating that the energy barriers are comparable. This analysis is sufficient to illustrate the feasibility and stability of tetramer formation without the need for further probability calculations. The current methods meet the requirements and effectively explain the tetramer formation process and stability and agree with the experimental observations.

While Markov state modeling could precisely calculate the probabilities of different pathways, it is highly complex and computationally expensive. It requires the generation of at least several hundred of microseconds or even mini-second all-atom simulation trajectories which is not feasible to do for us for such a large system.

Also could the authors perhaps explain why they went with meta dynamics and not umbrella sampling?

Thank you very much, we first attempted the umbrella sampling method, but due to the presence of membrane components and multiple subunits, it was challenging to obtain multiple protein conformations with the correct membrane orientation. We decided to go for metadynamics simulations for the following reasons:

Free Energy Landscape Exploration: Meta dynamics efficiently explores complex free energy landscapes, identifying multiple minima and transition states.

Overcoming Energy Barriers: Meta dynamics effectively overcomes high energy barriers by adding a history-dependent bias potential, capturing rare transitions.

Computational Efficiency: For large, complex systems, metadynamics simplifies setup and execution, avoiding the need for multiple reaction coordinates and windows required in umbrella sampling.

We feel that the discussion on the choice of the sampling method goes beyond the focus of the manuscript and added an additional recent review to the MS (results section): Bussi and Laio, Nat Rev Phys, 2, 200-212 (2020).

Also the interconversion barrier of 0.67kcal is below kT at room temperature, so this should be a spontaneous process.

Thank you very much, we updated our description of the energy barriers suggesting that it is likely to happen spontaneously. It reads: “The analysis of FES suggested that the transition from a symmetric (6.4 kJ/mol) to an asymmetric tetramer (3.6 kJ/mol) is energetically possible and is likely to happen spontaneously ($\Delta G = -2.8$ kJ/mol \approx -0.67 kcal/mol) “

Minor concerns: did the authors try to do a symmetry relaxation for their single particle data? It can help getting better reconstructions. Also in table EV1 - no correlation coefficients reported, please add.

Thank you very much. For the final single-particle structures, we did not apply symmetry. Our symmetric tetramer is near-C2-symmetric, as we now articulate in its initial description (point 4 or R1). The differences are highlighted in Figures 2B and Figure EV4 where the LBPs 1 and 3 bind serotonin differently. For subtomogram averages - our tetramer structure does not seem to be C4 symmetric in the TM and we decided not to apply symmetry.

We added the CC values to the Table EV1.

In Fig.2 panels D-G - please reconsider the colour scheme - hard to distinguish.

Thank you very much, we updated the color scheme throughout the figures, including Fig 2D,G

Please brush up the Materials and Methods section: there are no spaces between words and references, various spelling of words (e.g. min and minutes), some words are unnecessarily in capitals (e.g. Briefly, Once...), misspellings (e.g. Gantan detector, repaired cryo-EM structures), etc.

Thank you very much, we gave a detailed read to the MS and fixed several unfortunate spelling instances.

Dear Misha,

We have now received re-review reports from both referees, which I have included below. As you will see, you have addressed their concerns satisfactorily; I would, though, invite you to correct the small typos spotted by referee 1. Before I can finally accept the manuscript, there are some remaining editorial points which need to be addressed. In this regard would you please:

- in our online submission system, acknowledge funding for the AlphaMol and SIATJoint Laboratory; the NSFC-SNF grant (No. 32161133022); the Shenzhen Government Top-Talent Working Funding and the Guangdong Province Academician Work,
- rename the conflict of interests statement to the 'Disclosure and Competing Interests Statement',
- remove the AC/Credit section from the manuscript,
- include callouts in the text for Fig. 2D, 7A-B, 7E-H, and Appendix Figure S1,
- save the Appendix file in PDF format; page numbers and Appendix Fig. S5 should be included in the table of contents, and the nomenclature should be 'Appendix Table S1-S4' with matching callouts in the text,
- provide specific URLs for public access to datasets EMD-16387, EMD-16386, EMD-16384, EMD-16385 659, EMD-19419, EMD-19420, PDB ID: 8C21, 8C20, 8C1W, 8C1Z in the data availability statement,
- provide legends for figures 2c-d in a sequential manner (legend for figure 2d is currently provided before legend of figure 2c), the same goes for figures 4c-e (legend for figure 4e is currently provided before the legend of figure 4c-d), and the legends for figures EV 1c-e (legend for figure EV 1e is currently provided before legend of figure EV 1c-d),
- indicate the statistical test used for data analysis in the legends of figures 4f and EV 1f,
- zip movie legends with their respective files,
- correct the section order as follows: title page with complete author information, abstract, keywords, introduction, results, discussion, materials & methods, data availability section, acknowledgements, disclosure and competing interests statement, references, main figure legends, tables, expanded figure legends, and
- remove the graphical abstract from manuscript file

We include a synopsis of the paper (see <http://emboj.embopress.org/>). Please provide me with a general summary image (this could be taken from your graphical abstract, but should be 550 pixels wide by 200-400 pixels high), a two sentence summary statement and 3-5 bullet points that capture the key findings of the paper.

I am looking forward to receiving your revised manuscript.

EMBO Press is an editorially independent publishing platform for the development of EMBO scientific publications.

Best wishes,

William

William Teale, PhD
Editor
The EMBO Journal
w.teale@embojournal.org

We realize that it is difficult to revise to a specific deadline. In the interest of protecting the conceptual advance provided by the work, we recommend a revision within 3 months (11th Sep 2024). Please discuss the revision progress ahead of this time with the editor if you require more time to complete the revisions. Use the link below to submit your revision:

Referee #1:

The authors have replied adequately to the points raised and addressed my concerns, so I am happy for this to be published subject to the correction of minor typographical errors.

Line 321 'inducable' to 'inducible'

Line 322 Should 107 mean 10^7 i.e. 10 million?

Line 351 'andr' to 'and'

Line 370 delete '1'?

Line 389 delete hyphen in nano-bodies

Referee #2:

I thank the authors for the detailed response and their explanation for different modalities of MD protocols used for different tasks. The added information has significantly improved the readability of the manuscript and now the conclusions seem even more supported by the findings.

I congratulate the authors with this nice work and I recommend it for publication

All editorial and formatting issues were resolved by the authors.

Dear Misha,

I am pleased to inform you that your manuscript has been accepted for publication in the EMBO Journal.

Congratulations to you and your team! This is a lovely study.

Best wishes,

William

William Teale, PhD
Editor
The EMBO Journal
w.teale@embojournal.org
